# Multicentric tracking of multiple agents by anterior cingulate cortex during pursuit and evasion

Seng Bum Michael Yoo 🔟 1,2,3,4 ✉, Jiaxin Cindy Tu1,5 & Benjamin Yost Hayden 🔟 1

Successful pursuit and evasion require rapid and precise coordination of navigation with adaptive motor control. We hypothesize that the dorsal anterior cingulate cortex (dACC), which communicates bidirectionally with both the hippocampal complex and premotor/ motor areas, would serve a mapping role in this process. We recorded responses of dACC ensembles in two macaques performing a joystick-controlled continuous pursuit/evasion task. We find that dACC carries two sets of signals, (1) world-centric variables that together form a representation of the position and velocity of all relevant agents (self, prey, and predator) in the virtual world, and (2) avatar-centric variables, i.e. self-prey distance and angle. Both sets of variables are multiplexed within an overlapping set of neurons. Our results suggest that dACC may contribute to pursuit and evasion by computing and continuously updating a multicentric representation of the unfolding task state, and support the hypothesis that it plays a high-level abstract role in the control of behavior.

[1] Department of Neuroscience, Center for Magnetic Resonance Research, and Center for Neuroengineering, University of Minnesota, Minneapolis, MN, USA. [2] Center for Neuroscience Imaging Research, Institute for Basic Science, Suwon, Republic of Korea. [3] Department of Biomedical Engineering, Sungkyunkwan University, Suwon, Republic of Korea. [4] Present address: Department of Brain and Cognitive Sciences, Massachusetts Institution of Technology, Cambridge, MA, USA. [5] Present address: Department of Neuroscience, Washington University at St.Louis, St.Louis, MO, USA. ✉email: sbyoo.skku.bme@gmail.com

Foragers often encounter mobile prey that are capable of fleeing them. Not surprisingly, pursuit is a major element of the behavioral repertoires of many foragers[1–5]. Likewise, many foragers must also avoid predators seeking to capture them (e.g., ref. [6]). Recent studies have begun to identify the computational processes underlying pursuit and evasion behavior (hereafter shortened to pursuit) in several species[7,8]; nonetheless, the neural bases of these behaviors remain almost wholly unexplored (but see ref. [9]). Despite this relative paucity of scholarly interest, pursuit is an important problem in neuroscience because it is common in mobile animals and because it is highly determinative of reproductive success, and thus a likely driver of evolution. Moreover, it represents a mathematically tractable form of continuous decision-making, which decision neuroscience often ignores in favor of discrete decisions[10–12].

When foragers move in their environments, neurons in the hippocampus and adjacent structures track the forager's own positions of using a firing rate code[13–15]. They do so by means of an explicit map[16,17]. Specifically, each neuron in the hippocampus or medial entorhinal cortex exhibits one or more preferred firing fields[14]. That is to say, entry by an animal into a specific location results in robust spiking activity. The hippocampal spatial map is allocentric, meaning that it is organized relative to external space[13,18]. To employ this information to guide actions, however, foragers must use a complementary egocentric coding system, that is, one that is relative to the self[18–20]. Egocentric spatial representations are related to action planning and are often associated with the premotor cortex/primary motor cortex[21,22], and sometimes with the parietal and posteromedial cortex[22,23]. Even when navigating virtual or abstract environments, foragers can benefit from multiple reference frames. That is, they can make an abstract, allocentric or *world-centric* representation, but when the time comes to perform an action, or to navigate the virtual space, they may need to use an egocentric coordinate system, one that is aligned to the framework of their response modality. (Note that we will use the terms world-centric and avatar-centric below because our subjects are not performing a typical navigation task, but our hypotheses and our analyses are directly motivated by the existence of allocentric and egocentric mapping, respectively).

In addition to monitoring one's place in space, pursuit requires the careful coordination of two distinct processes: (1) the computation and dynamic updating of a representation of the pursuit environment, including the kinematics of the prey and predator; (2) the ability to select and quickly adjust behavior in response to changing demands. In other words, pursuit requires the coordination of cognitive mapping functions with motor control functions. To understand the cognitive mapping element of this process, we were especially interested in brain regions that have strong inputs from, on one hand, the hippocampal complex and, on the other, the premotor and motor system. The dorsal anterior cingulate cortex (dACC) fits this description[24–27]. Specifically, it is one of a small number of regions that receive converging information from reward regions (in this case, orbitofrontal cortex, amygdala, and insula) and navigational brain regions (parahippocampal and entorhinal cortices, and the hippocampal formation), and provides a direct output to motor brain regions, including the primary and supplementary motor cortices.

Some evidence supports the idea that both rodent and human dACC may carry place-relevant information, suggesting it may play a mapping role in pursuit[28–33]. We hypothesized dACC carries a rich and dynamic representation of key variables needed for pursuit decisions. Note that there is no a priori reason to assume that pursuit and evasion both rely on the same circuits. Indeed, it is likely that the neuroanatomy that mediates these two processes differs at least somewhat. Still, we reasoned that they

may have some overlap and that this overlap, if it exists, is may be most likely to occur within the dACC.

We recorded responses of neurons in the dACC of two macaques performing a real-time pursuit task. Subjects used a joystick to smoothly and rapidly move an avatar around a virtual pen on a computer screen to pursue fleeing prey and avoid predators that were chasing them. All agents other than the subject were controlled by interactive algorithms that used video game-derived artificial intelligence strategies. We found that dACC neurons track world-centric kinematic variables (specifically, position, velocity, and acceleration) for all three agents (self, prey, and predator). Although the responses of dACC neurons are spatially selective, they are more complex and multimodal than a place or grid cells would be (and in this, they resemble non-grid cells of the medial entorhinal cortex, see ref. [34]). Neurons in dACC also track the two key avatar-centric variables: relative position and angle of the self. Together, these results highlight the doubly framed role of dACC in monitoring complex relational positions, and provide a basis for understanding the neuroscience of pursuit and evasion.

## Results

**Pursuit and evasion behavior of macaques**. We measured responses from macaque dACC neuronal ensembles collected during a demanding computerized real-time *pursuit task* (subject K: 5594 trials; subject H: 2845 trials, "Methods"). A subset of these data was analyzed and summarized for a different study; all results presented here are new[33]. On each trial, subjects used a joystick to control the position of an avatar (a yellow or purple circle) moving smoothly in a rectangular field on a computer monitor (Fig. 1a–d and Supplementary Movie 1). Capture of prey (a fleeing colored square) yielded a juice reward delivered to the subject's mouth via a metal tube. The prey item on every trial was drawn randomly from a set of five that differed in maximum velocity and associated reward size. On 50% of trials (randomly determined) subjects had the opportunity to pursue either or both of two different prey items (but could only capture one). On 25% of trials (randomly determined), subjects also had to avoid one of five predators (a pursuing colored triangle). Capture by the predator ended the trial early, imposed a timeout penalty, and resulted in no reward.

Subjects successfully captured the prey in ~80% of trials (subject K: 78.95%; subject H: 84.91%). The average time for capturing a prey item was 3.85 s (subject K: 4.05; subject H: 3.50; Fig. 1e). To a first approximation, capture time did not differ according to the prey value/speed ($F = 50.98$, $p = 0.3797$ for subject K; $F = 26.68$, $p = 0.6118$ for subject H, two-way ANOVA). (This likely reflects a deliberate feature of the task design, which balanced value and speed to result in approximate equal capture times). On trials in which subjects faced two prey (50% of all trials), they had to choose which to pursue. On these trials, subjects chose the higher valued prey more often, even though those prey were faster and presumably more difficult to catch (K: 67.10%; H: 86.46%, Fig. 1f). Overall, these patterns suggest that subjects understood that prey color provided valid information about the value and/or speed of the prey, and used this information to guide behavior. This pattern also suggests that, for the parameters we chose, the marginal increase in reward value was more effective at influencing choice, on average, than the marginal increase in capture difficulty.

**World-centric encoding in dACC**. We recorded neuronal activity during performance of the task ($n = 167$ neurons; 119 in subject H and 48 in subject K). We applied a generalized linear model approach (GLM; refs. [34,35]) based on a linear–nonlinear

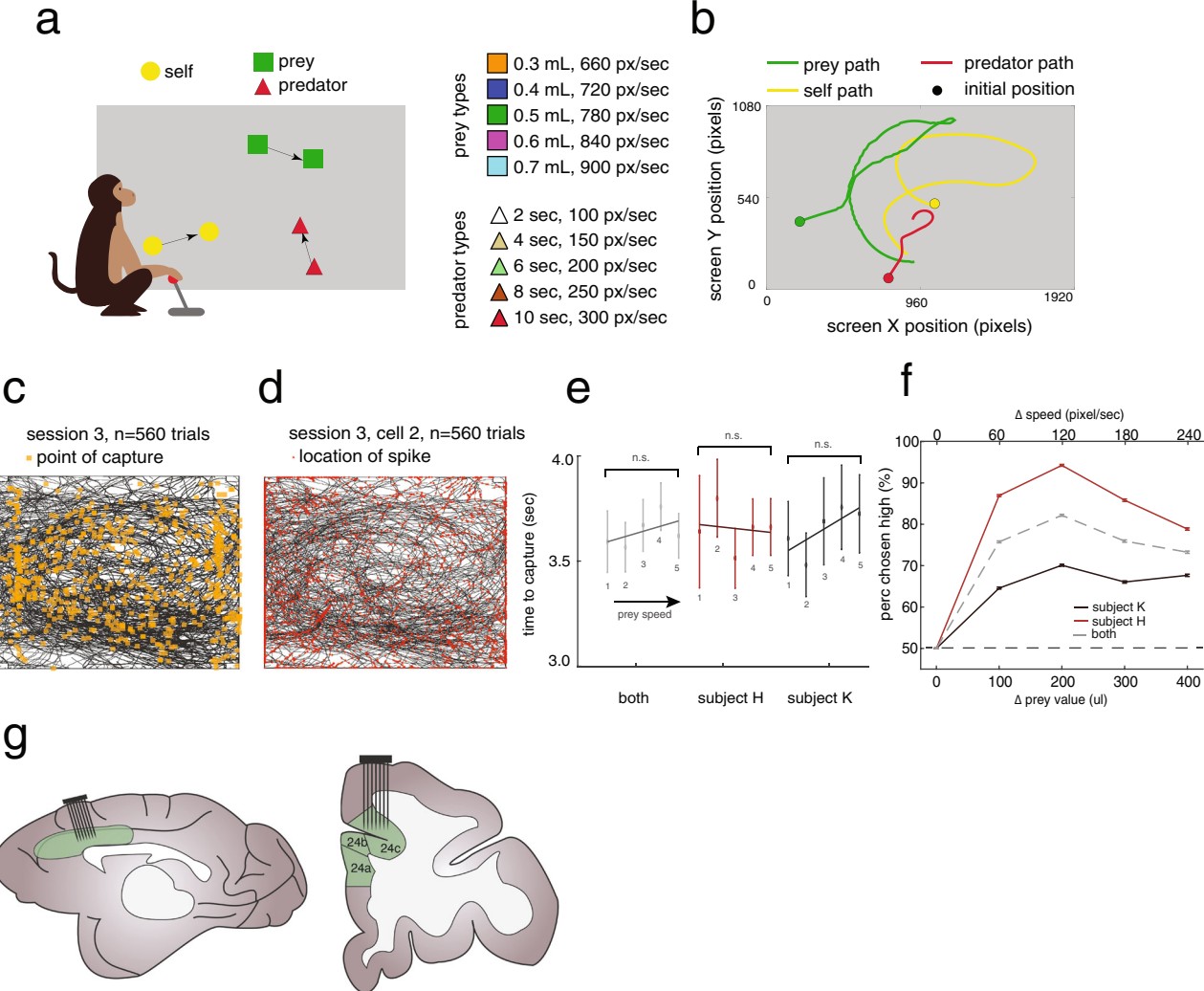

**Fig. 1 Experimental paradigm and behavioral results. a** Cartoon of the *virtual pursuit task*. Subject uses a joystick to control an avatar (circle) to pursue prey (square) and avoid predators (triangle). **b** Agent trajectories on example trial. **c** Illustration of typical session indicating subject's path (black lines) and points of capture (orange dots). **d** Illustration of example session showing subjects' path (gray lines) and points of spiking (red dots). **e** Average time to capture for each subject as a function of value of prey. No significant relationships are observed despite large datasets. This lack of correlation suggests that subjects generally traded-off effort to maintain roughly constant pursuit duration. **f** The proportion of trials on which subjects chose the larger value prey item despite it being faster was >50% for each prey difference (other than matched). Error bars indicate the standard error of the mean. The total number of trials in each subject was 5002 across 19 experimental sessions for subject K, and 3094 across 5 experimental sessions for subject H. **g** Schematic of the macaque brain showing medial surface and coronal sections. We record primarily from the dorsal bank of the anterior cingulate sulcus at roughly the rostrocaudal position of the genu; we call this region area 24.

(LN) model that does not assume any parametric shape of the tuning surface (see ref. [34], Fig. 2a, b). This procedure includes a cross-validation step, meaning that the results are essentially validated for statistical significance against a randomized version of the same data. This approach effectively includes a test for reliability, and also efficiently uses information about spatial coherence, to detect significant spatial selectivity. Note that, although we don't report the data, we confirmed that all results presented below are observed in both subjects individually.

Our analysis approach is a way of asking whether a neuron shows significant tuning (for example, whether it has angular tuning, Fig. 3a) but is agnostic about the shape of the tuning (for example, whether that place field is localized to a point, as hippocampal place fields are, or has a more complex shape). To identify the simplest model that best described neural spiking, we used a forward search procedure that determined whether adding variables significantly improved model performance. We used a

tenfold cross-validation to avoid overfitting (Fig. 3b–d). This cross-validation step ensures that reported effects reflect true measured patterns in the data.

For neural analyses, we focused on the *whole trial epoch*, that is, the period from the time when all agents appear on the screen (trial start) until the end of the trial, defined as either (i) the time when the subject captures the prey, (ii) the time when the predator captures the subject, or (iii) 20 s pass without either other event occurring. Thus, the tuning maps presented below indicate the average firing of neurons (using 16.67 ms analysis bins), during the entire course of active behavior during the trial, rather than during any specific epoch.

Examining this epoch, we found that 90.4% ($n = 151/167$) of neurons are task-driven, meaning that neuronal responses depend on one or more of the variables we tested (Fig. 3a, b, e, f). Note that the structure of our analysis, which forward searches for tuning for each variable automatically corrects for multiple

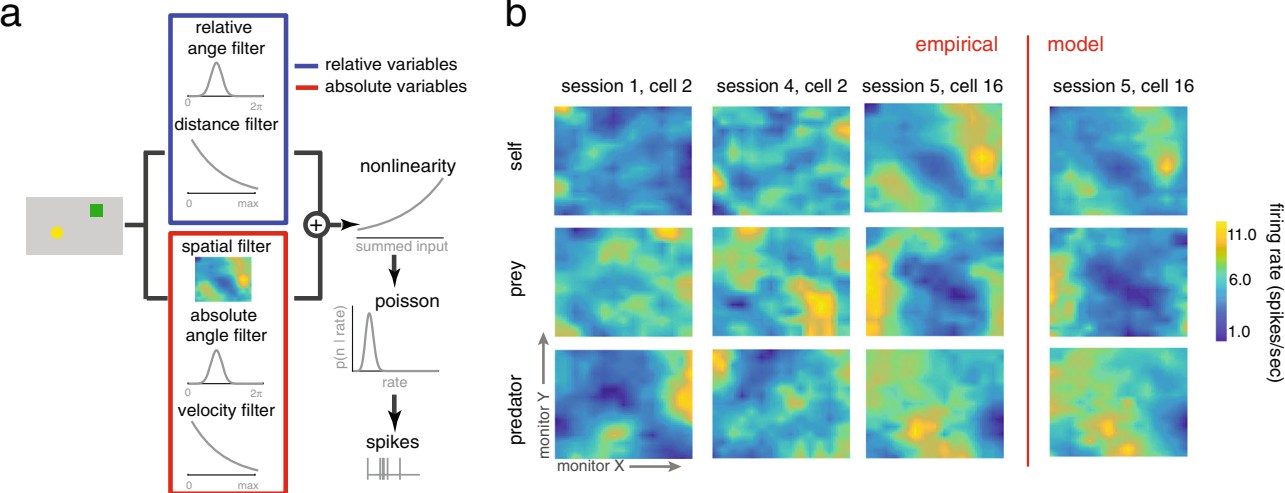

**Fig. 2 Examples of multi-agent world-centric mapping functions in dACC. a** Schematic of the analysis approach we took, see "Methods" and ref. [34]. **b** Maps of the responses of three example neurons with significant spatial selectivity, chosen to illustrate the typical types of responses we see. X- and Y-dimensions in each subplot correspond to X- and Y-dimensions of the computer monitor; the Z-dimension corresponds to the firing rate of that cell. Yellower areas indicate spots in which entry tended to result in spiking. Top row: responses to self position. Middle row: responses to prey position. Bottom row: responses to predator position. Rightmost column: responses of the model (see "Methods") for the same neuron whose observed responses are shown in the adjacent column.

comparisons (i.e., equivalent to using a cutoff of $p = 0.05$ for all variables).

The majority of neurons showed sensitivity to the spatial position of the avatar (64.5%, $n = 108/167$). Roughly similar proportions of cells showed sensitivity to the position of the prey (65.5%, $n = 112/167$), and to the position of the predator (59.3%, $n = 89/150$). We found that responses of 22.0% ($n = 33/150$) of neurons are selective for the positions of all three agents. Note the predator fits were done separately because they only occurred on 25% of trials, and for that analysis, we removed 17 cells that had trial counts below our a priori threshold (a number for predator trial <50). Note also that there is no certainty that subjects are engaging in pursuit and evasion simultaneously; indeed, it may well be the case that they alternate between these two modes, or blend them in different proportions at different times (see below).

We next tested whether overlapping populations of neurons encode self and prey position by examining log-likelihood increase (LLi) associated with adding the relevant variables (Fig. 3g). For each variable pair, we found a positive LLi relationship, indicating that neurons encoding one variable are more likely to encode the other, and therefore, evidence against specialized subpopulations of neurons for these variables. In other words, we found that populations overlap more than expected by chance (self/prey $r = 0.7882$; self/predator: $r = 0.7092$; prey/predator: $r = 0.6548$; $p < 0.001$ for all cases, Pearson correlation). This finding indicates that coding strength is positively correlated for each pair, and that the coding comes from a highly overlapping set of populations rather than from distinct subpopulations (see ref. [36] for motivation for this analysis approach). Thus, the two populations of neurons overlap more than might be expected by chance if these effects were distributed at random in the population. This result thus allows us to reject the hypothesis that the two groups of world-centric and self-centric neurons come from distinct sets—or even from overlapping sets that diverge more than might be expected by chance. Nonetheless, while these results are consistent with the idea that the neurons come from a single population, they are also consistent with the idea that they come from populations that overlap more than chance, but are still partially distinct (cf. ref. [37]).

We next used a previously published method to assess how the spatial kernels for the three agents compare (SPAtial Efficiency or SPAEF, see ref. [38]). For each pair of agents, we focused on neurons that show significant tuning for both agents individually. These groups consisted of, respectively, subject and prey 24.0%, $n = 36/150$; subject and predator: 26.0%, $n = 39/150$; prey and predator: 39.3%, $n = 59/150$. Incidentally, the largest of these three variables, perhaps surprisingly, was for the prey–predator. It's not clear why this is. One possibility is that this variable was encoded most strongly because of the special difficulty subject face in coordinating between pursuit and evasion strategies, and the need to attend to both other elements when doing so.

SPAEF is more robust than simple pairwise correlation because it combines three measures into a single value. Specifically, it combines pairwise correlation, coefficient of variation of spatial variability, and intersection between observed histogram and simulated histogram, see "Methods"). Across all neurons, we found that the SPAEF value between the subject and prey was −0.3282. This negative value indicates that the kernels are anticorrelated—locations that led to enhanced firing when the subject entered led to reduced firing at times when the prey entered it. This SPAEF value is significantly less than zero ($p < 0.001$, Wilcoxon sign-rank test). The value for the subject and predator was −0.2463. The analogous value for the prey and the predator was −0.2927. Both of these are also less than zero as well ($p < 0.001$, Wilcoxon sign-rank test). These findings indicate that neurons use distinct and anticorrelated spatial codes for tracking the positions of the three agents. These results suggest that dACC carries sufficient information for decoders to estimate path variables for all three agents.

We next asked whether a substantial number of neurons encode "self vs. other". To do this, we examined the set of neurons with significant selectivity for self position and for prey and/or predator position, and that showed a high positive predator–prey SPAEF value (that is, did not distinguish prey from predator). We found that six neurons meet these criteria (mean SPAEF value among those neurons is 0.2501). This proportion (3.6% of cells) is not significantly different from chance, suggesting that self vs. other encoding is not a major factor driving dACC responses.

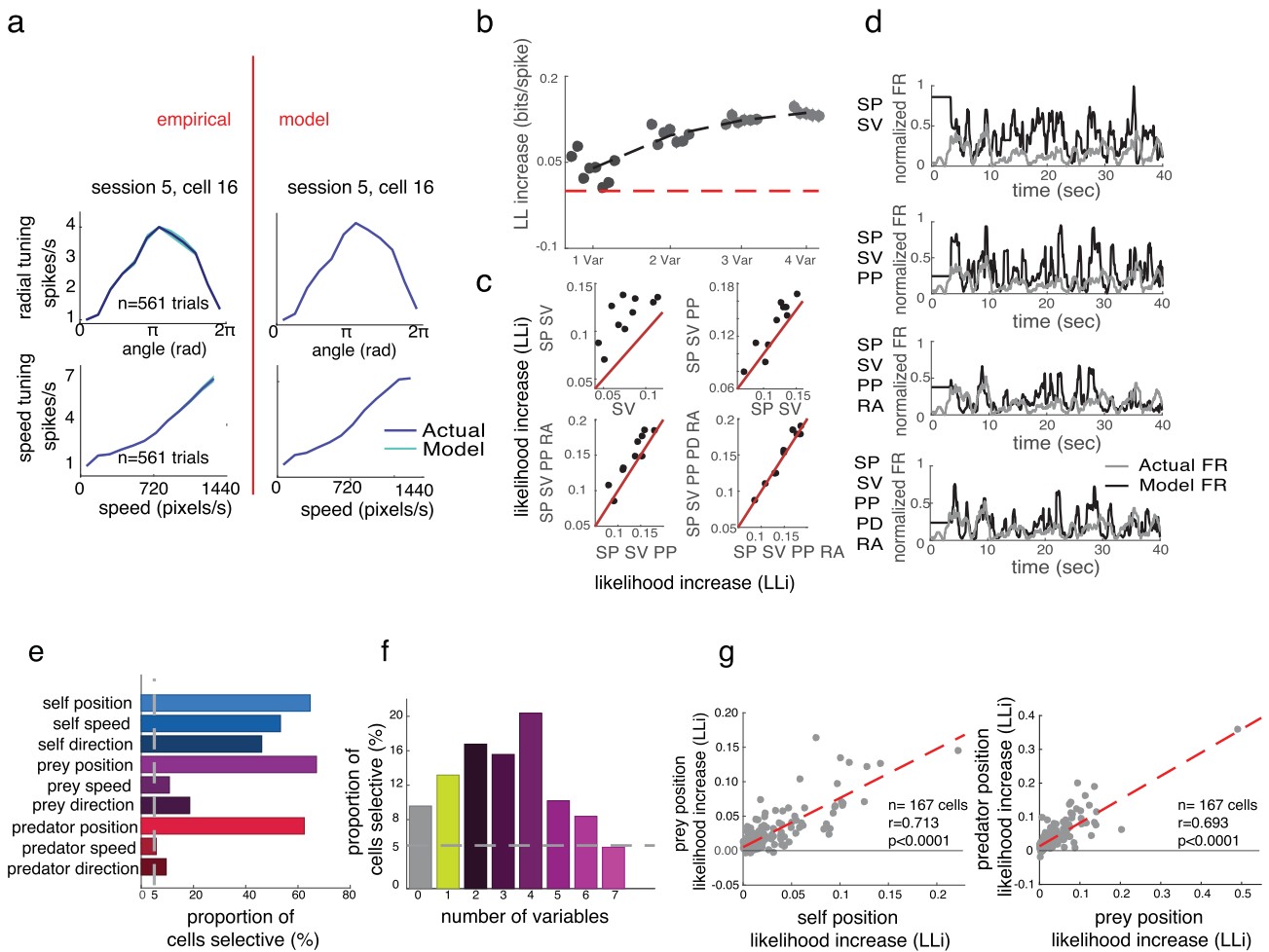

**Fig. 3 World-centric tracking in dACC neurons. a** Direction and speed tuning (left) and model fit (right) for an example neuron. The error bands are connection of standard deviation in each data point. Each error bars are standard deviation for 1000 times of bootstrapping the neurons. **b** Proportion of neurons selective for 1, 2, 3, or 4 of the world-centric variables self position, self velocity, prey position, and relative angle between self and prey. Horizontal jitter added for clarity. Red dashed horizontal line indicates the likelihood value for mean firing rate model, which we treated as a zero variable model. **c** Likelihood increase in fit as a function of adding additional variables. This method was used to select variables via cross-validation. The significance of tuning with respect to the variable was determined by sign-rank test between red unity line and tenfold cross-validation data points. **d** Example neuron showing observed firing rate (gray) and model predicted firing rate (black). The held-out data are plotted. **e** Proportion of neurons tuned for each possible number of task variables. Proportions do not sum to 1 because neurons may be tuned for multiple variables; results reflect correction for multiple comparisons. **f** Proportion of neurons in our sample selective for any of 7 number of variables. Most neurons are selective for multiple of the variables we test. **g** Correlation between log-likelihood increase (LLi) for self position vs. prey position (left) and prey position vs. predator position (right). Each dot corresponds to one neuron. Positive correlation indicates selectivity is a shared property of selective cells. The significance is obtained from Pearson correlation between LLi of specific variables.

Our results reveal spatial maps but do not indicate their raw size in terms of firing rate, and thus risk overstating extremely minor influences on firing that are nonetheless significant. We thus next sought to characterize the size of these effects. To do so, we first selected neurons that showed significant selectivity for the position of each agent. Then, for each neuron in each set, we selected the peak firing rate and lowest firing rate in the two-dimensional (2D) space. This measure is analogous to peak-to-trough measures. We then computed the median within each set. (Median is more conservative than mean because it more effectively excludes outlier measurements, which visual inspection revealed to be a modest risk). We find substantial effects for each category; self-tuned neurons: 13.34 spike/s (95% confidence interval: 9.44–17.24 spike/s); prey-tuned neurons: 11.77 spike/s (95% confidence interval: 7.65–15.89 spike/s); predator-tuned neurons: 12.55 spike/s (95% confidence interval: 7.21–17.09 spike/s). These effects

are quite large, and are comparable to modulations associated other factors in more conventional laboratory tasks.

**Speed information is also processed in dorsal anterior cingulate cortex.** We hypothesized that dACC would encode agent speed. To test that idea, we added speed filters in our GLM and fit against the neural data. We found that 22.7% ($n = 34/150$) neurons are selective for the speed of the self, 10.0% ($n = 15/150$) of neurons for the speed of the prey, and 10.0% ($n = 15/150$) for the speed of the predator (Fig. 3a). Naturalistic tasks such as ours provide the opportunity to understand higher-dimensional tuning than other methods. To gain insight into the diversity of speed tuning profiles, we performed an unsupervised $k$-means clustering on speed filters across the agents (Fig. 4). Initially, we performed principal component analysis (PCA) on the filter coefficients. Then, we obtained eigenvector of top two dimension, in which explained >70% of variance of the data. We find both

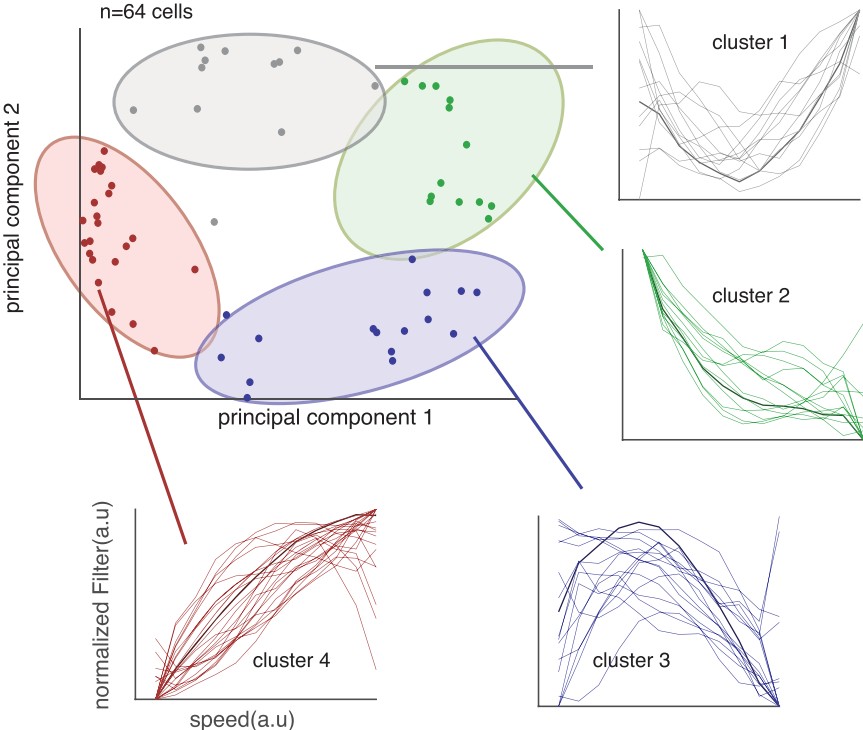

**Fig. 4 Diversity in responses of neurons to speed.** We examined whether speed tuning reflects a single response profile, as would be expected if, for example, speed effects were simply an artifact of arousal. We performed a PCA procedure on all tuning curves and plot the results as a function of each PC. We then cluster the resulting patterns and show all tuning curves within each cluster. The diversity of responses, and especially the existence of clearly ditonic clusters (clusters 1 and 3), argues against an arousal confound.

monotonically and ditonic speed filters (11 neurons for cluster 1, 16 neurons for cluster 3). Previous literature suggested that better perceptual discrimination for lower speed[39]. Interestingly, our result shows that neurons in ACC, at least, is not biased toward representing low speed.

**Avatar-centric encoding.** We next examined avatar-centric coding, that is, coding of the position of the prey relative to the agent. According to our GLM, 37.7% of neurons ($n = 63/167$) in our sample encode the distance between self and prey, and 25.1% of neurons ($n = 42/167$) encode the angle between self and prey (Fig. 5a). Together, these two variables define the entire basis set of avatar-centric spatial variables relevant to the pursuit of the prey. That is, other avatar-centric variable can be expressed as a linear combination of them, and thus are available to decoders that have access to the responses of these neurons. A smaller proportion of neurons signal these variables relative to the predator ($n = 14/150$, 9.3% for relative distance to predator, $n = 8/150$, 5.3% for relative angle to predator). Note that the value for relative distance to predator is significant, while the value for angle is not (distance: $p = 0.0220$; angle: $p = 0.8496$; one-way binomial test; Fig. 5b, c). Nonetheless, in toto, these results indicate that dACC neurons carry a rich representation of the avatar-centric world in this task.

We were concerned that distance tuning, as determined by this analysis, may be artifactual—it may reflect proximity to reward, which is known to consistently enhance activity in dACC[40,41]. To test this alternative hypothesis, we performed an analysis of the diversity of responding. Specifically, we reasoned that if neurons encode distance, they will show a heterogeneity in response patterns but if they encode proximity to reward, they will show a more homogeneous and positive-going pattern. To examine our hypothesis, we clustered the shape of subject–prey distance filters (Fig. 5d). These figures use the following radial plot conventions.

The angle on the plot relative to 0 (i.e., horizontal and to the right) reflects the angle between the subject's own avatar and the prey. Thus, a neuron selective for the subject bearing directly toward the avatar will have lighter colors on the right-hand side of the radial plot. The radial dimension on the plot indicates the distance—thus, a neuron selective for distant prey will have lighter colors on the outer ring of the plot.

We observed a heterogeneity of curves, including a substantial fraction of neurons with decreasing and even ditonic curves (48.3%, $n = 29/60$, $p < 0.001$, two-way binomial tests). The ditonicity (i.e., positive and negative slopes within a single curve) of some neurons is important—it indicates that these neurons do not simply exhibit ramping behavior. This result thus argues against the possibility of avatar-centric distance simply being an artifact of the proximity of reward, and/or arousal or other low-level features that scale with distance to reward (Fig. 5d).

By identifying avatar-centric-coding neurons, we were able to ascertain whether avatar- and world-centric-coding neurons arose from different or similar populations. We used the same log-likelihood correlation approach described above. We find that they are not distinct; instead, they overlapped considerably more than would be expected by chance (that is, the correlation of log-likelihoods was greater than zero, $r = 0.295$, $p < 0.001$; Fig. 5e, f). This result is consistent with the possibility that these neurons come from a single task-selective population, as well as with the possibility that they come from highly overlapping, but partly distinct sets.

**Mixed selectivity.** Encoded variables interacting nonlinearly (mixed selectivity) is potentially diagnostic of control processes and can be harnessed for flexible responding[34,42,43]. We used two methods to test for mixed selectivity ("Methods" and Fig. S3). First, we computed direction and speed tuning separately in high and low firing rate conditions (a method found in ref. [44], Fig. 6a).

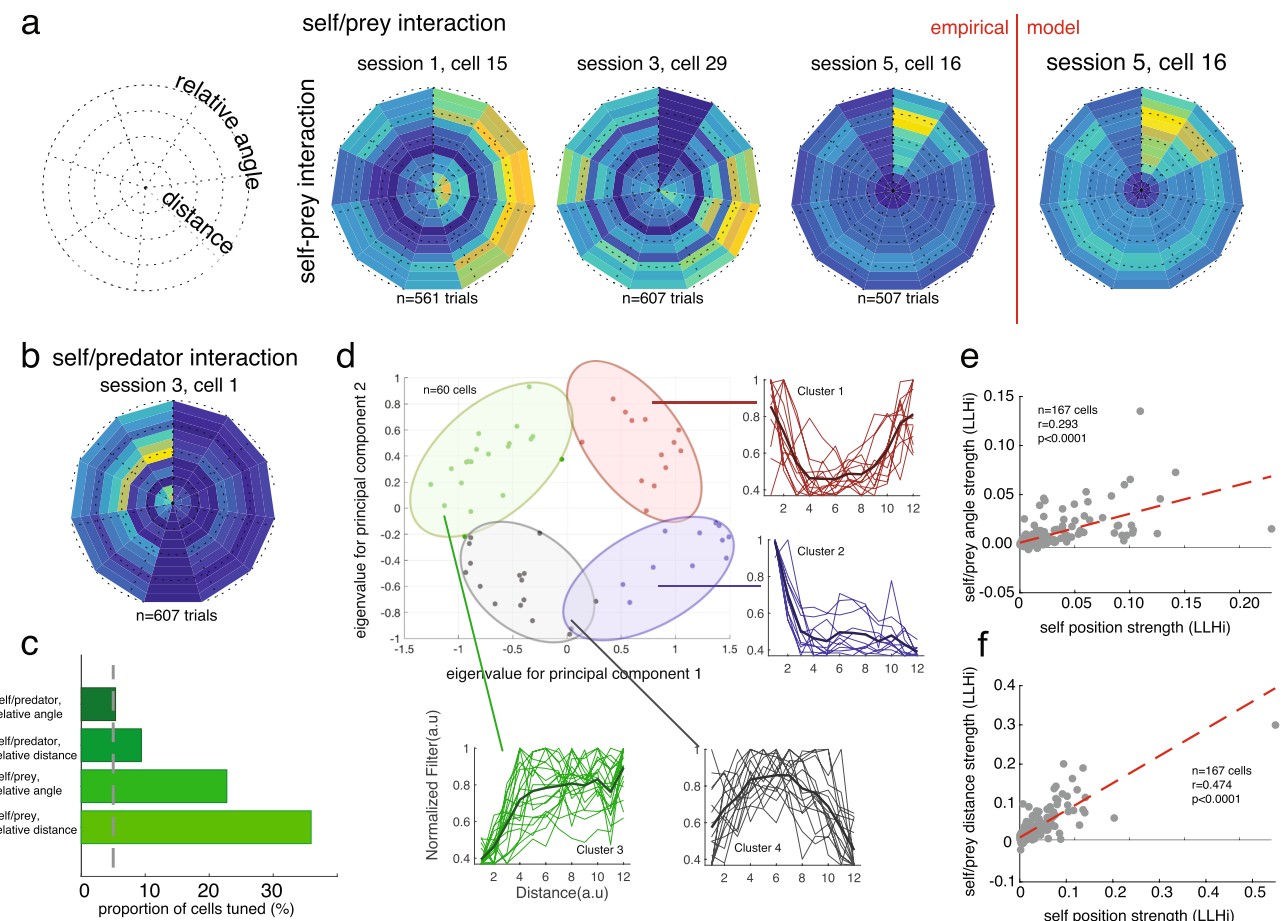

**Fig. 5 Avatar-centric tuning in dACC. a** Three example neurons and model fit for self-prey avatar-centric variables (angle and distance). Each radial plot shows the responses of a single neuron to prey located at several distances (radial dimension) and angles (angular dimension) relative to the agent's avatar. Horizontal right direction (0°) reflects distance in front of the direction the avatar is currently traveling; horizontal left (180°) indicates travel away from the prey. **b** Example neuron tuned for self-predator avatar-centric variables using same conventions as in **a**. **c** Proportion of neurons tuned for each of four avatar-centric variables. **d** Clusters of relative distance tuning functions (conventions as in Fig. 4). Scattergram reflects results of principal components analysis (PCA). Clusters 3 and 4 are ditonic; their existence argues against low-level explanation in terms of arousal or reward proximity. **e**, **f** Correlation between log-likelihood increase (LLi) for self position vs. self/prey distance (**e**) and angle (**f**). Each dot corresponds to one neuron. Positive correlation indicates that neurons selective for one variable tend to be more selective for another one. That in turn implies that tuning for the two variables comes from a single larger population of cells (or from highly overlapping populations) rather than distinct populations.

Then we performed regressions for the two conditions separately. A slope different from 1 indicates a multiplicative shift; an offset different from 0 indicates an additive shift. In our data, the median of the slope was significantly <1 (median slope = 0.8533, $p < 0.022$, rank-sum test) with little evidence for additive modulation (position: median bias = 0.4909, $p < 0.001$, rank-sum test; speed: median multiplicative factor (MF) = 0.7692, $p = 0.048$; median additive factor (AF) = 0.4661, $p < 0.001$; Fig. 6b, c). This result indicates that dACC ensembles have the capacity to represent information in high-dimensional space by encoding multiple variables nonlinearly[42,45].

We confirmed this mixed-selectivity result with an additional method that is less sensitive to the shape of the tuning curve[34]. Specifically, we characterized the range (max firing rate–min firing rate) of each tuning curve, as a function of the mean firing rate for the position (three bins; method from ref. [34]). As expected under mixed selectivity, the range increased with mean position segment firing rate (median $r = 0.2305$, $p < 0.001$, rank-sum test; Fig. S3). Together these two results indicate that dACC neurons use nonlinearly mixed selectivity (and not just multiplexing) to encode various movement-related variables.

**Spatial coding is distributed across neurons.** Although responses of a large number of neurons are selective for spatial information about the three agents, it is not clear to what extent a broad population drives behavior[46]. Thus, we examined how much each neuron in the population contributes to behavior using population decoding with an additive method[47]. If only small sets of neurons contribute to behavior, the decoding performance with respect to number of neurons will soon reach a plateau. We randomly assigned neurons to the decoder regardless of whether they were significantly tuned to the variable of interest. We found that as the number of neurons included for decoding analysis increases, the accuracy of decoding positional variable (both self and prey) increases without evidence of saturation (Fig. 7a).

We next applied this serial decoding procedure to examine relative strength of different formats of spatial coding. For this analysis, we focused on coding of world-centric angle (self and prey direction) and avatar-centric angle, which share common units (specifically, degrees; Fig. 7b). We find that the strength of information within the neural population is mixed between world-centric and avatar-centric information. Self-direction information is strongest and prey direction information is

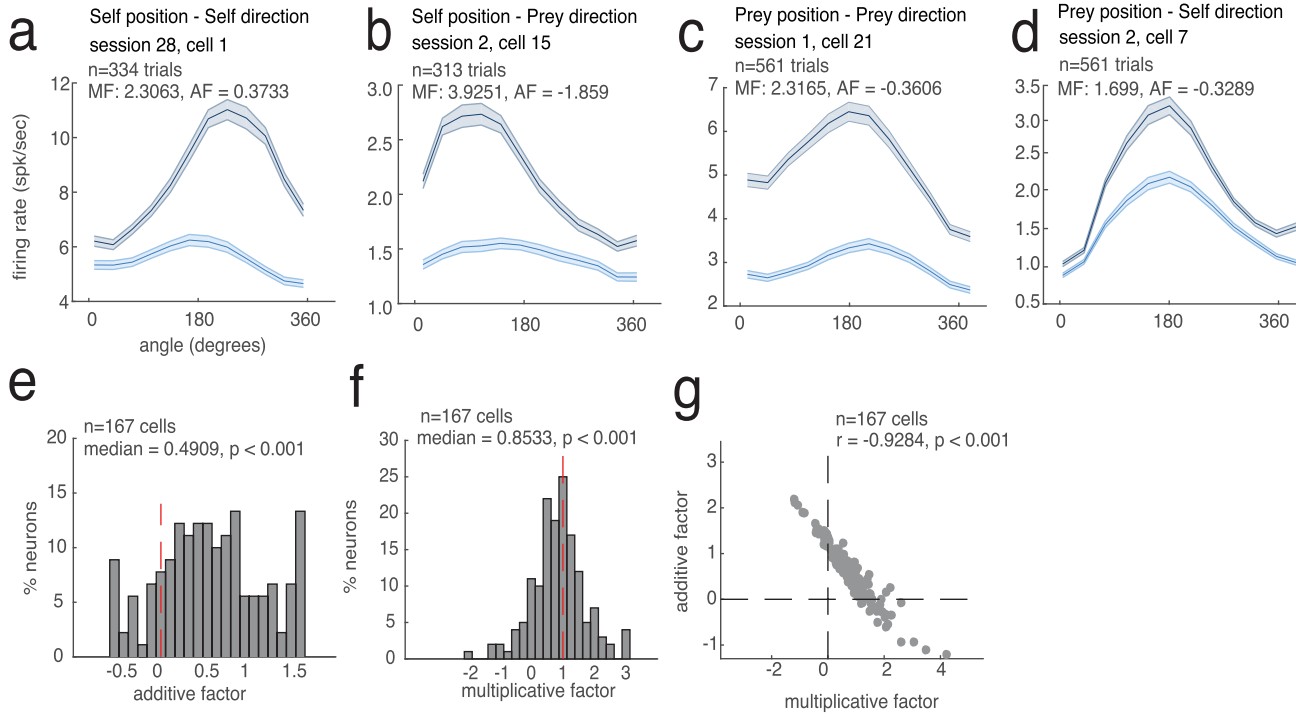

**Fig. 6 Mixed selectivity.** We illustrate our findings on mixed selectivity by focusing on relative angle. See supplement for other variables. **a–d** Angular tuning curves for example neurons that showed significant mixed selectivity. Horizontal axes reflect angle between self and prey; 0 indicates heading toward the prey. Neurons show angular selectivity that is modulated by anticipated reward. Gray line: high-value prey. Blue line: low-value prey. **a** Multiplicative influence of self direction by self position, **b** self direction by prey position, c prey direction by self position, and **d** prey direction by prey position. The error bands are connection of standard deviation in each data point. Each error bars are standard deviation for 1000 times of bootstrapping the neurons. **e**, **f** Additive (**e**) and multiplicative (**f**) factors of population. AF > 0 indicates additive interaction; MF deviates from 1 indicates multiplicative interaction. **g** Relationship between MF and AF. Negative correlation indicates that neurons with multiplicative interaction are less likely to have additive interaction.

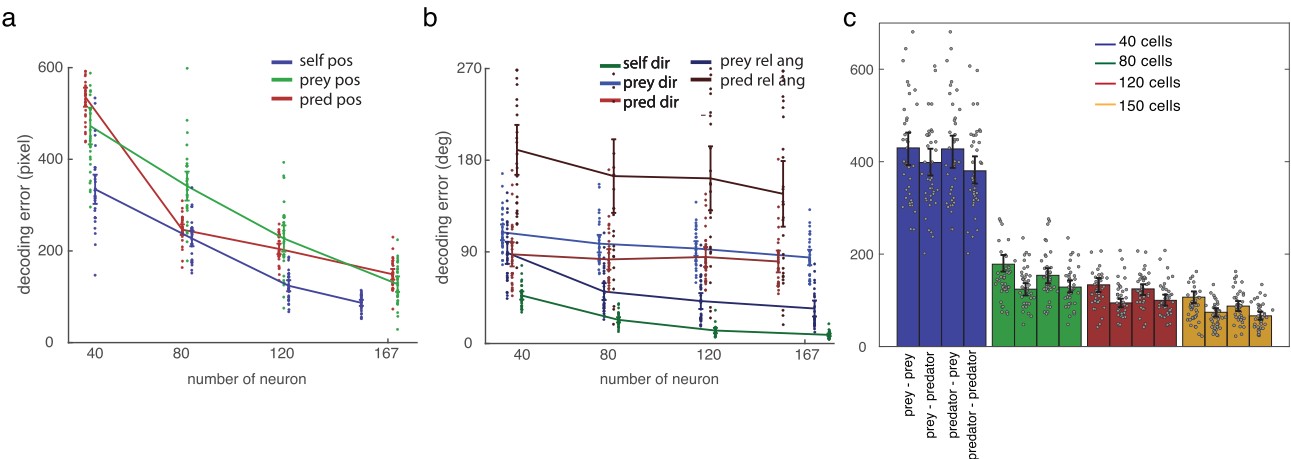

**Fig. 7 Decoding performance of the model. a** Population decoding accuracy (distance between model and empirical data) for self (blue) position, prey (green), and predator (red) position improves (i.e., gets lower) as number of neurons analyzed rises. This analysis indicates that these variables are encoded in a distributed manner, and that all three agents can be readily decoded. **b** Population decoding accuracy for self (dark green) direction, prey (purple) direction, predator (red) direction, prey relative angle (dark blue), and predator relative angle (maroon). All curves decrease modestly, indicative of distributed decoding. **c** Time-to-time decoding performance when either prey is closer or predator is closer. Then, the decoding performance for either prey position or predator position was estimated. Decoding accuracy was estimated as a function of number of cells, in groups corresponding to 40, 80, 120, and 150 cells. Statistical significance and error bar widths (i.e., SD) are obtained from bootstrapping the neural population for 30 times (total $N = 167$ neurons).

weakest (and their difference is significant, decoding error by using all neurons are $8.496 \pm 0.603°$, $34.364 \pm 3.795°$, $84.455 \pm 3.712°$, $p < 0.001$, ANOVA). This result showing distributed information contrasts with previous findings in a similar paradigm that show positional variables are encoded by only a handful of neurons[46]. We speculate that difference may due to the complexity of our task, which may require a high-dimensional neural space to maximize the information[48].

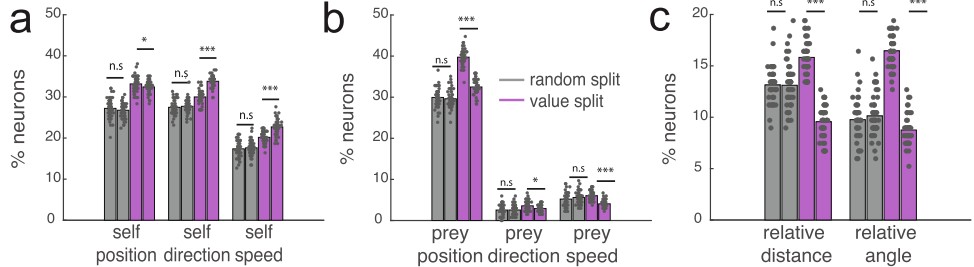

**Fig. 8 Tuning for task variables change with prospective reward.** These data support the idea of mixed selectivity. **a** The number of tuned neurons for self variables (self position, self direction, and self speed) when splitting data randomly (gray bar) or according to value of pursued prey (purple bar). Splitting by value increases proportion of tuned neurons, indicating that value modulates responding in a systematic way. **b** The number of significantly tuned neurons for prey variables (prey position, prey direction, and prey speed) when splitting data randomly (gray bar) or according to value of pursued prey (purple bar). The difference of value split was significant ($p = 0.0221$ for prey speed, and $p < 0.001$ for other prey variables), but not for random split. For the prey analysis, only single prey trial is included to reduce confounding (i.e., pursuing targets, transition of pursuit-avoidance behavior). **c** The number of significantly tuned neurons for egocentric (self position, self direction, and self speed) when splitting data randomly (gray bar) or according to value of pursued prey (purple bar). The difference of value split was significant ($p < 0.001$). We observed a difference in numbers of significantly tuned neurons between split data. This process was repeated for 50 times by bootstrapping. Significance was calculated by two-sided sign-rank test. n.s means it is not significant, and *** indicates $p < 0.001$.

**Reward encoding**. Research based on conventional choice tasks indicates that dACC neurons track values of potential rewards[49]. We next asked how dACC encodes anticipated rewards in our more complex task. Initially, we regressed reward variable against the neural activity 1 s before the trial end for all types of trial. We found that, averaging over all other variables, the value of the pursued reward modulates activity of 9.3% of neurons ($n = 14/150$, $p = 0.0227$, one-way binomial test). Note that this analysis ignores the potential encoding of prey speed, which is perfectly correlated with static reward in our task design. We then explored possibility of reward being modulatory variable, which means that reward increase the other variables' selectivity. We find that tuning for all variables increases with increasing reward ($p < 0.05$ in each case, sign-rank test, Fig. 8). Compare to the random split of data, which yielded insignificance difference in tuning, splitting data according to the value of prey did yield a significant difference in tuning proportions for the variables (Fig. 8). Importantly, the percent of neurons tuned for each variable is maintained in the random split, indicate reliability of tuning. Instead, the proportion of neurons whose responses were selective for self position was not different, when the data were split randomly into half (28.0% vs. 27.4%, $p = 0.2783$, sign-rank test, 50 times bootstrapping).

**Gaze does not change selectivity of spatial tuning**. Activity in dACC is selective for saccadic direction and may therefore also correlate with gaze direction[50]. Consequently, it is possible that our spatial kernels may reflect not task state but gaze information. In the dynamic pursuit task, the position of the eyes is not fixed. Indeed, subjects continually scan the scene and follow specific items on screen. This leads to the possibility of a novel confound—specifically, our "world-centric" representations may come from simple gaze direction tuning. This would not necessarily invalidate our claims because gaze-centrism may be a mechanism, by which world-centric coding is enacted within the context of our task. However, the data we have suggests that the world-centric encoding we see is largely independent of gaze coding.

The strongest evidence that the mapping functions we observe are unlikely to reflect a gaze confound is that we observe anticorrelated tuning surfaces for the self and prey and for the self and predator (see the section showing SPAEF results). If selectivity was derived from gaze selectivity, these surfaces would necessarily be identical. Moreover, we repeated our GLM

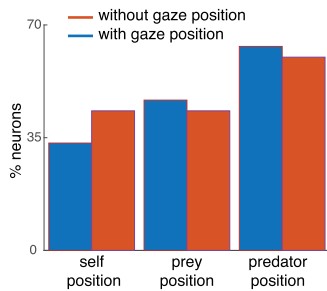

**Fig. 9 Analysis to detect potential gaze confounds.** Red bars: proportion of neurons tuned for three key world-centric variables using the standard GLM described above. Blue bars: same results, but this time from a version of the GLM that included eye position as a regressor. That version is constructed so that all variance possible associated with eye position is assigned to eye position first and only residual encoding of task variables is counted toward those variables. All three variables are still significantly observed in the population when including gaze position. Significance was tested by two-way binomial test ($n = 10/37$, $p < 0.001$).

analyses, but included eye position (only for the one subject from which we collected gaze data). We found that neurons are selective for gaze position (confirming past research on the issue), but that this selectivity was largely independent of the selectivity for the self, prey, and predator. Specifically, we found that that after including gaze, the proportion of tuned neurons for the position of any agent did not substantially change (Fig. 9).

To shed further light on these issues, we also offer some characterization of saccade behavior in this task. One question related to its free eye movement is in what proportion of the time subjects foveate each agent (self, prey, and predator). To gain the broadest view, we analyzed one-prey and one-predator trials, wherein all types (subject, prey, and predator) of agent exist (7008 trials in two subjects). Our criterion for defining foveation was a range of four-degree visual angle, which corresponded to 134 pixels (for reference, each agent was 60 pixel for width and height). We found that the foveation time is highest for the prey (31.85% and 19.05% for subjects K and P, respectively), and lowest for the predator (19.09% and 5.11%, respectively), with a small proportion for foveating both prey and predator (2.82% for subject K, 0.07% for subject P). Foveation on the avatar occurs 24.34% of the time (subject K) and 9.71% for subject P.

A second question is how eye movements relate to target pursuit. One possibility is that they maximally use fixations/ saccades. In other words, they may position their fovea around the predicted position of the target and fixate until the tracking target moves beyond some angle. An alternative possibility is to follow the target with smooth pursuit. (These two strategies have some heuristic relationship with discrete and continuous sampling, respectively). We found that the latter hypothesis corresponds to our data. Specifically, 90.88% (subject K) and 94.70% (subject P) of time bins have eye movements that are smooth pursuit, 4.03%/3.11% are saccadic fixation, and 5.10%/ 2.18% are movement.

We hypothesized that the appearance of the saccade may be related to the complexity of the pursuit trajectory (Fig. S4). We used the curvature of the agent movement as a proxy for the complexity of trajectory, and measured cross-correlation between curvature value and magnitude of eye movement. By this analysis, we wanted to answer two questions: (1) what is the nature of the saccade pattern? (Does large eye movement correlate with large curvature/) and (2) is there any systematic time lag between the saccade and the start of complex movement pattern? Indeed, we found that there is a systematic relationship between the complexity of the prey (i.e., moment-to-moment curvature of the trajectory) and the saccadic eye movements (magnitude of the eye movement pixelwise). About 50% of the trials showed a significant cross-correlation between those two values (median correlation: $p = 0.5020$, $n = 2052/3954$ trial for subject K; $p = 0.5334$, $n = 1395/3054$ trials for subject P; $p < 0.001$ by two-way binomial test for both subjects). The median time lag was $-66.7$ ms for subject K and $-116.7$ ms for subject P. The large eye movement as leading the high curvature, which indicates the saccade (global scanning of the environment) happens before the pursuit becomes complex.

Finally, we asked whether eye position information influences the tuning of other variables. Specifically, we asked two questions: (1) whether the tuning for other non-gaze-related variables is decreased when the gaze-related variables are added, (2) whether the gaze variables are not significantly tuned when they are added to the fits because the correlated variables already captured variance. We found neither is the case: the tuning of other neurons remains significant even after adding gaze position information (Fig. 9). If the correlation really influences the tuning, the tuning of prey position should be decreased for the largest amount as the foveation time is highest at the prey. However, tuning for the prey did not decrease significantly. In addition, a significant number of neurons are tuned to the gaze position ($n = 10/37$, $p < 0.001$ by two-way binomial test), which means there is variance still well captured by the gaze position information.

**Classification of behavioral strategies**. To more fully understand our subjects' behavior, we created multiple artificial agents that pursue prey and avoid predators. We used these agents to estimate the efficiency of the monkeys' observed algorithm (Fig. S5). There are two components that vary across our five models: (1) the predictive parameter tau, which indicates how much extrapolation from previous Newtonian physics that an agent can make toward the future (Yoo et al.[12,33]), and (2) the distance-dependent influence of the predator. If the distance-dependent function for the predator is large, it influences the subject even if the predator is far from it. All of these algorithms are bound to the same physical constraints, especially to a maximum speed and physical inertia. We divided the tau parameter into two categories (tau was either predictive or reactive) and the influence parameter into two categories (low distance/narrow or high distance/broad

influence of predator). We then crosses these to make four categories and added a final degenerate random walk model.

These models give a range of performances. The *random walk model* gives 0% capture of prey within 20 s (not surprisingly; Fig. S5). One trials with predators, actual subjects catch prey on 54.05% (subject K), and 38.97% (subject H) of trials. For this analysis, we also include data from a new subject whose data did not appear in our earlier submission (subject P), whose performance was between the other two (45.55%). Other agents that always predictively pursue (fixed tau value = 30) the prey yields a higher catch rate compared (86.54% for narrow attention for predator vs. 75.03% for broad attention for predator). However, if the agent is always reactive (fixed tau value of $-30$, which differs from subject's average trajectory), then though the agent is faster than the prey, it rarely catches the prey. Instead, the probability of being captured by a predator increases (10.15% for narrow attention for predator, 0.13% for broad attention for predator). In conclusion, the predictive model with narrow attention is the most accurate descriptor.

**Moment-by-moment chase and avoidance behavior**. We estimated the avoid and chase moments by basic statistics given each moment-by-moment (Fig. S6). Included variables are change of the distance (delta distance: "getting close or far?") and the dot product between the vectors of each agent ("how similar the movements are") and its change ('are the agents moving similarly over time?'). For example, if the delta distance, dot product, and delta dot product is positive between subject and predator (example in figure), it will be more likely for the mode of avoidance. According to this method, clear avoidance times were 10.48% (subject K), 9.09% (subject H), and 12.99% (subject P) from the whole session.

## Discussion

We examined the neural foundations of pursuit and evasion by recording single unit activity in the dACC, while rhesus macaques performed a joystick-controlled pursuit task (Fig. 1). We find that dACC carries a dynamic (i.e., continuously updated) multicentric (i.e., both world-centric and avatar-centric) multi-agent (self, prey, and predator) representation of the state of the task. These results indicate a clear role of the dACC in mapping functions that are intrinsic to pursuit. One limitation of the present study is that we did not record activity in other regions that may also be involved. Therefore, we cannot conclude that dACC plays a unique role in this process. Future studies will be needed to functionally differentiate dACC from other regions.

What is the benefit of encoding both absolute (world-centric) and relative (avatar-centric) maps? One possibility is that dACC participates in the process of mediating between the two representations. Another (not mutually exclusive) possibility is that both representations are important for behavior. Consider, for example, that avatar-centric codes may allow for rapid on-the-fly changes in trajectory, while world-centric ones may allow for more abstract planning, for example, allowing the subject to trap the prey in corners. Having both in the same place may allow for their coordination to make optimal decisions. Indeed, this idea is consistent with the idea that a major function of dACC is to use multiple sources of information to set and drive a strategy from a high vantage point[24–26,40,41].

Neuroscientists are just beginning to understand the neural basis of tracking of other agents. Traditionally, neurons in primate dACC and its putative rodent homologues are not expected to encode place fields. For example, a putative rodent homologue is reported to utilize positional information, but not signal place per se[28]. However, at least one notable recent study has

demonstrated that place field information can be decoded from rodent ACC[46]. Our results build on this finding and extend our understanding of the spatial selectivity of this region further to tracking of other agents. Two recent studies demonstrate the existence of coding for positions of conspecifics in the CA1 region of the hippocampus in rats and bats[51,52]. Our results here extend on them in three ways. First, they confirm speculation that positional tracking extends to at least one hippocampal target region in the prefrontal cortex. Second, they demonstrate that positional tracking extends to multiple agents, including different types (prey and predators), and that it is multicentric. Most intriguingly, and most speculatively, our results directly link tracking of others to personal goal selection processes.

Although the responses we observe have some similarity to hippocampal place cell firing, dACC responses are less narrowly localized than place cells, are less patterned than grid cells, and can only be detected using a newly developed statistical approach[34]. While the medial entorhinal cortex is commonly associated with grid cells[53–55], one recent study demonstrated that it carries a much richer set of spatial representations[34]. Our study indicates that such noncanonical spatially mapped neurons are not limited to entorhinal cortex, or to rodents, and can be observed in virtual/computerized environments, and extend to other agents in the environment. These results confirm the highly embodied role of dACC in economic choice and highlight the central role of spatial information in economic decision-making[10,11,56–58].

Our data are limited to a single region and do not imply a unique role for this region. Most notably, several other ostensibly neuroeconomic brain regions carry rich spatial repertoires, including OFC[59] and vmPFC[60,61]. These regions also have connectivity that includes, directly or indirectly, medial temporal navigation regions, and motor and premotor regions. Therefore, we predict that the patterns we observe here would also be observed, albeit perhaps more weakly, in these other regions. Unlike these regions, the dACC has been linked to motor functions, albeit much less directly than, for example, motor cortex[25]. What is new here, then, is the observation that dACC tracks the kinematics of self, prey, and predator, uses multicentric tuning for these multiple agents. In addition to what it tells us about dACC, the multicentric multi-agent tuning also serves as a control for possible motor effects explaining the results.

Most studies of the neural basis of decision-making focus on simple and abstract choices, but natural decisions take place in a richer and more complex world. In our task, decisions are continuous—they take place in an extended time domain and the effects of decisions are manifest immediately. Moreover, our task, and monkeys' ability to perform it well, illustrate the complexity of the word decision—it has a simple and clear definition in economic choice tasks. But in a more naturalistic context, like this one, it can refer either to the specific direction the subject is moving at a point in time, or to the higher level goal of the subject. Ultimately, we anticipate that consideration of more complex tasks may lead to a refinement of the concept of decision.

More broadly, given the critical role of foraging in shaping our behavioral repertoires overall[62–64], we and others have proposed that spatial representations are likely to be a ubiquitous feature of our reward and decision-making systems[57,65]. This idea is supported, at the most basic level, by studies showing clear spatial selectivity in the reward system in both rodents and primates[59,60,66–69]. In other words, spatial information is not abstracted away even in ostensibly value-specialized regions[57,70]. By utilizing ever more complex paradigms, we can place the brain into natural states not probed by conventional tasks and uncover unanticipated complexities in neuronal responses.

## Methods

All animal procedures were approved by the University Committee on Animal Resources at the University of Rochester or the Institutional Animal Care and Use Committee at the University of Minnesota. All experiments were designed and conducted in compliance with the Public Health Service's Guide for the Care and Use of Animals.

**Subjects**. Two male rhesus macaques (*Macaca mulatta*) served as subjects. Note that the inclusion of only two subjects (as is standard in primate physiology) does reduce our ability to draw inferences about group effects. Previous training history for these subjects included a variant of a Wisconsin card-sorting task (subject H, ref. [71]), basic decision-making tasks (subject H, ref. [72]), and two foraging tasks (both subjects, refs. [73,74]).

**Experimental apparatus**. The joystick was a modified version of a commercially available joystick with a built-in potentiometer (Logitech Extreme Pro 3D). The control bar was removed and replaced with a control stick (15 cm, plastic dowel) topped with a 2″ plastic sphere, which was custom designed through trial and error to be ergonomically easy for macaques to manipulate. The joystick position was read out in MATLAB running on the stimulus control computer.

**Eyetracking methods**. Subjects were rigidly head-fixed using a cranial implant that was located far from the eyes. Subjects were facing directly forward with the eyes centered on the center of the computer monitor. Gaze position was measured with an Eyelink 1000 system at 1000 Hz. Calibration was performed daily for each subject using a 20-point calibration procedure, in which they fixated briefly on a point displayed at random on a computer screen. Calibration quality was monitored throughout the recording session, and was checked periodically during recording by pausing the task and running a calibration routine again. Careful post hoc checks reveal little systematic error in tracking; subjects were found to be placing gaze on the targets. These tests concorded with casual observations made during and throughout testing by a trainee monitoring the subject and gaze at all times. For resulting analyses, the gaze $x$ and $y$ position were included in analyses.

**Task design**. At the beginning of each trial, two or three shapes appeared on a gray computer monitor placed directly in front of the macaque subject (1920 × 1080 resolution). The yellow (subject K) or purple (subject H) circle (15-pixel diameter) was an avatar for the subject and moved with joystick position. A square shape (30-pixel length) represented the prey. The movement of the prey determined by a simple algorithm (see below). Successful capture is defined as any overlap between the avatar circle and the prey square. Each trial ends with either successful capture of the prey or after 20 s, whichever comes first. Capture results in immediate juice reward; juice amount corresponds to prey color: orange (0.3 mL), blue (0.4 mL), green (0.5 mL), violet (0.6 mL), and cyan (0.7 mL). Failure to capture results in timeout and a new trial. (Failures were rare).

The path of the prey was computed interactively using A-star pathfinding methods, which are commonly used in video gaming[75]. For every frame (16.67 ms), we computed the cost of 15 possible future positions the prey could move to in the next time step. These 15 positions were spaced equally on the circumference of a circle centered on the prey's current position, with radius equal to the maximum distance the prey could travel within one time step. The cost in turn is computed based on two factors: the position in the field and the position of the subject's avatar. The field that the prey moves in has a built-in bias for cost, which makes the prey more likely to move toward the center (Fig. 1a). The cost due to distance from the subject's avatar is transformed using a sigmoidal function: the cost becomes zero beyond a certain distance so that the prey does not move, and the cost becomes greater as distance from the subject's avatar decreases. The position with the lowest cost is selected for the next movement. If the next movement is beyond the screen range, then the position with the second lowest cost is selected, and so on.

The maximum speed of the subject was set to be 23 pixels per frame (i.e., 16.67 ms). The maximum and minimum speeds of the prey varied across subjects and were set by the experimenter to obtain a large number of trials (Fig. 1). Specifically, speeds were selected so that subjects could capture prey on >85% of trials; these values were modified using a staircase method. If subjects missed the prey three times consecutively, then the speed of the all prey was reduced temporarily. The minimum initial distance between the subject avatar and prey was 400 pixels. The strict correlation between speed and value means that value cannot be directly deconfounded in this study.

A predator (triangle shape) appeared on 25% of trials. Capture by the predator led to a timeout. Predators came in five different types (indicated by color) indicating different level of punishment, ranging from 2 to 10 s. The algorithm of the predator is to minimize the distance between itself and player. Unlike the prey, the predator algorithm is governed by this single rule.

The design of the task reflects primarily the desire to have a rich and variegated virtual world with opportunities for choices at multiple levels that is neither trivially simple nor overly complex. The decision to include a condition with multiple prey was added specifically for these reasons and for the additional reason that we wanted to verify that subjects distinguished the differently valued prey by pursuing them with differential preference.

The reason we deliberately confounded reward and speed was to make sure the task neither too difficult nor too easy, and to ensure that the results of the animals' choices between prey were interesting and meaningful. We also wanted to keep the effort/interest level roughly the same on each trial.

**Trajectory-based trial sorting**. In 50% of trials, subjects saw two prey items instead of one. We developed the TBTS (trajectory-based trial sorting) method to determine which prey the subject was pursuing at any given time. This method requires to calculate (1) the angle differences between subject's and each prey's trajectory from time $t$-1 to $t$, (2) change of distance between subject and each prey (estimate whether prey is getting closer to subject or floating), and (3) dynamic time warping outcome (to calculate the distance between the signal) between the trajectory of the subject and each prey. Then we multiply them to obtain a single scalar for each agent at every time point, and then smoothed with a boxcar of five frames to make secure autocorrelation between the data points. The prey being pursued will have smaller angle difference with the subject, the distance between the subject and pursued prey will be decreasing (due to avoiding algorithm, non-chased prey will tend to increase its distance with the subject), and dynamic time wrapping outcome will be smaller. Thus, when one prey is pursued continuously, then this value will stay always smaller than the other. From there, we excluded trials with switches to avoid any confounds that arose from not knowing what prey the subjects are pursuing (~5% of trials overall).

**Electrophysiological recording**. One subject was implanted with multiple floating microelectrode arrays (FMAs, Microprobes for Life Sciences, Gaithersburg, MD) in the dACC. Each FMA had 32 electrodes (impedance 0.5 MOhm, 70% Pt, 30% Ir) of various lengths to reach different dACC. Neurons from another subject were recorded with laminar V-probe (Plexon, Inc, Dallas, TX) that had 24 contact points with 150 μm inter-contact distance. Continuous, wideband neural signals were amplified, digitized at 40 kHz and stored using the Grapevine Data Acquisition System (Ripple, Inc., Salt Lake City, UT). Spike sorting was done manually offline (Plexon Offline Sorter). Spike sorting was performed blind to any experimental conditions to avoid bias.

**Tracking neurons over multiple days**. We used an open-source MATLAB package "Tracking neurons over multiple days"[76]. Briefly, pairwise cross-correlo-grams, the autocorrelogram, waveform shape, and mean firing rate were used together as identifying features of a neuron. For classifying the identical neurons across the session, we calculated four values that characterize individual neuron: mean firing rate, autocorrelation, pairwise correlation with other neurons, and shape of the waveform. Then, we applied a quadratic classifier that computes an optimal decision boundary under the assumption that the underlying data can be modeled as a mixture of multivariate Gaussians[76].

**Details of LN model**. To test the selectivity of neurons for various experimental variables, we adapted LN Poisson models (LN models). The LN models estimated the spike rate ($r_i$) of one neuron during time bin $t$ as an exponential function of the sum of the relevant value of each variable at time $t$ projected onto a corresponding set of parameters ($w_i$). The LN models can be expressed as:

$$r = \exp\left(\Sigma X_i^T w_i\right)/dt \qquad (1)$$

Where $r$ denotes a vector of firing rates for one neuron over $T$ time points, $i$ indexes the variables of interest, e.g., position of avatar on screen. $X_i$ is a matrix where each column represents an "state" of the animal (e.g., 1 of 12 speeds, determined by post hoc binning) obtained from binning the continuous variable so that all the columns for a particular row is 0 except for one column (one-hot encoding).

Unlike conventional tuning curve analysis, GLM analysis does not assume the parametric shape of the tuning curve a priori. Instead, the parameter weights, which defines the shape of tuning for each neuron, were optimized by maximizing the Poisson log-likelihood of the observed spike train given the model expected spike number ($n$), with additional regularization for the smoothness for parameters in a continuous variable and a lasso regularization for parameters in a discrete variable. Position parameters are smoothed across rows and columns separately. The regularization hyperparameter was chosen with maximizing cross-validation log-likelihood based on several randomly selected neurons. The optimization was performed with a MATLAB built-in function (fminunc). Model performance of each neuron is quantified by the log-likelihood of held-out data under the model. This cross-validation procedure was repeated ten times and overfitting was penalized. Thus, we can compare performance of models with varying complexity.

**Forward model selection**. Model selection was based on the cross-validated log-likelihood value for each model. We first fitted $n$ models with a single variable, where $n$ is the total number of variables. The best single model was determined by the largest increase in spike-normalized log-likelihood from the null model (i.e., the model with a single parameter representing the mean firing rate: $r$). Then, addi-tional variables ($n$-1 in total) were added to the best single model. The best double model was preferred over the single model only if it significantly improves the

cross-validation log-likelihood (Wilcoxon signed-rank test, $\alpha = 0.05$). Likewise, the procedure was continued for the three-variable model and beyond if adding more variables significantly improved model performance, and the best simplest model was selected. The cell was determined to be not tuned to any of the variables considered if the LLi was not significantly higher than baseline.

**Response profile**. We derive response profiles from filter of model for a given variable j to be analogous to a tuning curve of given variable. These were computed as, which $\alpha = \Pi i_{[i \text{ is all other variables than } j]}$ mean($\exp(w_i)$) is a scaling factor that marginalizes out the effect of the other variables. The d$t$ transforms the units from bins to seconds. Thus, for each experimental variable, the exponential of the parameter vector that converts animal state vectors into firing rate contributions is proportional to a response profile; it is a function across all bins for that variable and is analogous to a tuning curve.

**Principal component analysis and clustering relative distance tuning**. We reasoned that relative distance between subject and prey is encoded and is not simply an artifact of proximity to reward acquisition. This variable is encoded in dACC, although generally with robustly positive and monotonic code[73,74,77]. Instead, complex shape of tuning may indicate distance is encoded. To examine this, we clustered the tuning curves according to shape to show whether there might be some functional clusters. First, we selected out 60 neurons that are individually significantly tuned to relative distance of prey. Then, we performed dimensionality reduction via PCA and found two PCs explain 70% of variance in the data. Thus, we projected data into first two PCs and performed $k$-nearest neighbor and found elbow with $K = 4$ (Fig. 5d). Identical method was used for profiling the filters that are tuned for speed (Fig. 4).

**Multiplicative vs. additive shift of tuning**. We report that neurons exhibit "multiplicative" tuning, defined as $r(x, y) = r(x)^* r(y)$, which means the tunings for each variable interact nonlinearly, and thus have mixed selectivity[45]. However, there is possibility that the neurons might show additive tuning, defined as $r(x, y) = r(x) + r(y)$. Strictly, linear addition would be multiplexed but not mixed selectivity[42].

Differentiating between these two has important implications as multiplicative coding may point to a fundamental transformation of information, while additive coding suggests signals simply linearly combine[34,45].To quantify the nature of conjunctive coding and verify our assumption that tuning curves multiply, we examined neurons that significantly encoded both position of two agents (self and prey) and direction of two agents based on model performance (e.g., both the position and other models had to perform significantly better than a mean firing rate model). We examined differences in how the tuning curve for specific variables $r(x, y)$, or the tuning curve across $y$ for a fixed value $x^*$, will change as a function of $r(x^*)$ to estimate whether neurons exhibit multiplicative or additive. In the multiplicative model, a variation of $r(x^*)$ will modify the shape of tuning curve (either stretch or compress) $r(x^*, y)$, whereas in the additive model it will shift $r(x^*, y)$ simply up or down. To quantify these differences, we took $x$ to be position and $y$ to be either direction or speed of agent, and binned position into 15 × 15 bins. We then calculated the firing rate for each position bin (i.e., computed $r(x^*)$ for every $x^*$), sorted the position bins according to firing rate values, and divided the bins into two (high vs. low, for analysis 1, see below) or three (for analysis 2, Fig. S3) segments. Each segment, thus, corresponded to a location of the environment with approximately the same firing rate. We then generated a series of tuning curves (either direction or speed) based on the spikes and directions visited during each segment.

Once we obtained tuning curves for each segment, for each single neuron, we characterized its multiplicative, additive, or displacement modulation with population activity by performing linear regression on the average response to each state bins, when population activity was high compared with when it was low. The slope of the linear fit indicates how tuning scales multiplicatively with population activity (so called, MF). The slope deviating from 1 shows either multiplicative or displacement interaction. The intercept of the fit describes the additive shift to tuning with population activity. To obtain a relative measure of the additive shift, like the MF, we defined the AF as the ratio between this intercept and the mean firing rate of the neuron averaged.

We additionally confirmed the multiplicative tuning shift by computing the range (maximum firing rate–minimum firing rate) of tuning curve as a function of the mean firing rate for position segment $i$. If each neuron shows multiplicative tuning shift, the range should increase with position segment. Whereas the additive model result in constant range. The range of the tuning curve and mean position segment firing rate exhibited a positive slope in pool of significantly tuned neurons for self direction, self speed, prey direction and prey speed (134/165 pooled neurons; median slope > 0 with $p < 0.001$, two-way binomial test).

**Downsampling for reward modulation analysis**. Each session was split into low vs. high reward of pursued target. To match the coverage of experimental variable for each condition, we first binned position, direction, speed into 225 bins, 12 bins, and 12 bins, respectively, and computed the occupancy time for each bin. The coverage of experimental variables across conditions was matched by

downsampling data points from either condition so that occupancy time was matched for each bin, with points removed based on the difference in direction occupancy. For this analysis, we wanted to have the total number of spikes match in the compared conditions. We did this because spikes convey information and we did not want to introduce spurious differences in information content that reflected only random variation in number of spikes. We then repeated this procedure 50 times and took the average result. We did this because the decimation procedure is stochastic and there susceptible to random effects. We can reduce these effects and obtain a more precise estimate of the true effect, through repetition and averaging.

**Decoding analysis**. Decoding accuracy was assessed by simulating spike trains, which were based on Poisson statistics and a time-varying rate parameter. In each group, spikes ($n_c$) for neuron c were generated by drawing from a Poisson process with rate $r_c$; where ($w_{i,c}$ are the learned filter parameters from the selected model for neuron c, $X_i$ is the behavioral state vector, and $i$ denotes the experimental variables). If the model selection procedure determined that a neuron did not significantly encode variable $i$, then $w_{i,c} = 0$. Next, the simulated spikes were used to estimate the most likely variable that is being decoded. To decode experimental variables at each time point $t$ under each decoder, we estimated the animal state that maximized the summed log-likelihood of the observed simulated spikes from $t$-L to $t$:

$$X^{(t)} = \text{argmax}_t \sum_{l=0}^{L} \sum_{c=1}^{C=N} \log P\left(n_c \middle| \exp\left(\Sigma^{(t-l)^T} w_{i,c}\right)\right) \quad (2)$$

where $C$ is the number of cells in that population. Decoding was performed on 50 randomly selected 2000 ms ($L = 121$ time bins) of session. The average position decoding error (pixel distance error), direction (error in degree), and speed (error in pixel per second) were recorded. For examining the coding scheme of the population (whether sparse or distributed code), we increased number of neurons being included in this analysis from 40 to 167 by 40 neurons. The random shuffling of neurons was performed 30 times.

**Adaptive smoothing method**. An adaptive smoothing method is used for presentation purposes although not for quantified data analysis[78]. Briefly, the data were first binned into $100 \times 1$ vector of angle bins covering the whole 360° of the field, and then the firing rate at each point in this vector was calculated by expanding a circle around the point until the following criteria was met:

$$N_{\text{spikes}} \geq \frac{a}{N_{\text{occ}}^2 \times r^2} \quad (3)$$

where $N_{\text{occ}}$ is the number of occupancy samples, $N_{\text{spikes}}$ is the number of spikes emitted within the circle, $r$ is the radius of the circle in bins, and alpha is a scaling parameter set to be 10,000 as previous studies.

**Spatial efficiency metric**. To compare the similarity between two positional filters, we used the SPAEF metric. This is a mathematical technique that is derived from the geology literature, but has a much broader application. Formally, it allows for the quantification of the similarity of two 2D filters with univariate scalars as entries. Prior literature suggests to be more robust than the 2D spatial correlation (Koch et al.[38]). It quantifies the similarity between two maps:

$$\text{SPAEF} = 1 - \sqrt{(A-1)^2 + (B-1)^2 + (C-1)^2} \quad (4)$$

Here, $A$ is the Pearson correlation between two maps, $B$ is the ratio between the coefficients of variation for each map, and $C$ is the activity similarity measured by histogram profiles. A zero SPAEF indicates orthogonal filters, whereas a positive SPAEF indicates similar filters and a negative SPAEF indicates anticorrelated filters.

**Statistics and reproducibility**. The data of subject K are obtained from 19 repeated experiments, and the data of subject H are obtained from 5 repeated experiments. The data of subject P are obtained from 20 sessions to characterize the eye movement specifically.

**Reporting summary**. Further information on research design is available in the Nature Research Reporting Summary linked to this article.

## Data availability
The data will be available once manuscript is accepted. The full data will be shared upon the request to the corresponding author. The partial data to replicate figures will be uploaded at the GitHub. Source data are provided with this paper.

## Code availability
A portion of the data is available on Github (https://github.com/sbyoo/multicentric/). Full data are available from the corresponding author upon reasonable request.

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

## Acknowledgements
We thank Alex Thomé for his critical role in designing the task, for devising the training protocols, and for developing our joysticks. We thank Marc Mancarella for his critical help with joystick training. We are grateful for helpful discussions from Habiba Azab, Steve Piantadosi, Marc Schieber, and Adam Rouse. National Institute on Drug Abuse Grant R01 DA038106 (to B.Y.H.).

## Author contributions
S.B.M.Y. and B.Y.H. conceptualized and designed the experiment. S.B.M.Y. collected the data. S.B.M.Y., J.C.T., and B.Y.H. developed the model and analyzed the data. S.B.M.Y. and B.Y.H. wrote the manuscript.

## Competing interests
The authors declare no competing interests.
