## [Peer Review File · Nature Communications]

Reviewers' Comments:

Reviewer #1:

Remarks to the Author:

In this manuscript, Yoo et al. investigated neuronal coding for moving agents in the dorsal anterior cingulate cortex (dACC) of the monkeys. The authors trained the monkeys to perform the task in which the animals were required to control an avatar (self) to pursuit prey and to escape from a predator on a monitor using a joystick. They found that a set of dACC neurons showed significant tuning for 'world-centric' variables such as the positions and velocities of the agents. Also, a set of dACC neurons showed significant tuning for 'avatar-centric' variables such as the distance and angle between the self and prey agents. The topic of this study is timely and interesting for readers with a broad background. However, there are one major and several minor concerns that should be addressed.

Spatial coding of hippocampal place cells is thought to be 'allocentric' because activities of place cells are mostly tuned for the absolute coordinates of the environment and are independent of the sensory-motor information such as head-direction and locomotion speed. In the current study, the authors claimed that the populations of dACC neurons represent 'world-centric' spatial representations of the agents. Here, the 'world-centric' is used as the similar meaning of 'allocentric.' However, they did not show clearly whether the world-centric representations were only based on objective variables of the agents or also dependent on sensory-motor signals such as gazing. This is a critical point for discussing world-centric representations of neurons. The analysis of Figure 9 seems insufficient to clarify the contributions of gazing. Firstly, the basic analyses about the relationships between the gazing point and the agent (self, prey, predator) position will be needed. How much the gazing point correlates to the position of each agent? How often and with what types of behavior relations did the saccades occur? Secondly, neuronal tuning for only gazing variables should also be analyzed. (This kind of data is missing in Figure 9.) Although they mentioned the influence of gazing and attention (Line 472-484), more thorough analysis and discussion seem to be needed.

Minor comments

- 1) Please provide the details of the analysis using spatial efficiency (SPAEF) in Methods.
- 2) Please provide the information on the experimental methods of eye-tracking in Methods.
- 3) Related to Figure 9. Please provide the information on what gazing variables were used in the analysis.
- 4) Line 861~. What is the time window for fitting the spike trains to the model?

Reviewer #2:

Remarks to the Author:

Yoo SBM et al., Multicentric tracking of multiple agents by anterior cingulate cortex during pursuit and evasion

The authors present data from a study in which they recorded neural activity from the dACC while animals carried out a task in which they pursued prey objects and avoided predators. They carried out a detailed receptive field analysis and found that neurons encoded predator, prey and self positions, as well as direction from self to prey and predator.

Overall, I found that this paper addressed an interesting problem. The introduction framed the experiments nicely within a broader ethological and anatomical framework. The task was well-designed with nice features and the data set was large. Analyses were detailed and thorough. I have

only a few comments/ suggestions with one major comment.

Comments

1. The main thing I thought the study was missing was a detailed analysis of what the animal was actually doing. This could then form the basis of the neural analysis, which were descriptive and detailed, but could be more insightful. Perhaps the authors have tried some of these approaches. One possibility would be to analyze the direction of motion of the subject, moment by moment, and determine the extent to which it is driven more by pursuit of prey or avoidance of predators? This likely depends on the size of the reward, and the proximity of the predator. I realize this only occurs in a subset of trials. But it could be used to divide the neural activity into epochs, or look for some sort of switching phenomenon, as in switching between pursuit/avoidance. Also, how efficient was the prey pursuit? One could imagine something between a random walk and a predictive pursuit strategy, that tries to intersect the prey based on its current direction and speed.

2. A better figure indicating recording locations would be useful. Is this 9m? 24c? Medial area 6? Dorsal to the cingulate sulcus? This may be in the methods, but it would be useful to show locations and maybe given approximate AP coordinates.

3. The manuscript was well-organized but could definitely benefit from editing. Results are often presented in the present tense (are instead of were) and there are numerous typos.

Reviewer #3:

None

Reviewer #1 (Remarks to the Author):

In this manuscript, Yoo et al. investigated neuronal coding for moving agents in the dorsal anterior cingulate cortex (dACC) of the monkeys. The authors trained the monkeys to perform the task in which the animals were required to control an avatar (self) to pursue prey and to escape from a predator on a monitor using a joystick. They found that a set of dACC neurons showed significant tuning for 'world-centric' variables such as the positions and velocities of the agents. Also, a set of dACC neurons showed significant tuning for 'avatar-centric' variables such as the distance and angle between the self and prey agents. The topic of this study is timely and interesting for readers with a broad background. However, there are one major and several minor concerns that should be addressed.

We appreciate these positive comments. We have now addressed all comments listed; we believe that the resulting changes have made the paper better. Detailed replies to the specific comments are found below.

Spatial coding of hippocampal place cells is thought to be 'allocentric' because activities of place cells are mostly tuned for the absolute coordinates of the environment and are independent of the sensory-motor information such as head-direction and locomotion speed. In the current study, the authors claimed that the populations of dACC neurons represent 'world-centric' spatial representations of the agents. Here, the 'world-centric' is used as the similar meaning of 'allocentric.'

Indeed, the reviewer is correct about this. We apologize for any confusion, and have added the recommended terminology in hopes of making things less confusing. Specifically we now explicitly add the terms "allocentric" and "egocentric" as comparators in the Introduction. However, we still preferentially use the original terms throughout so as to not to imply that these neurons are tested using a standard maze/navigation-like task.

However, they did not show clearly whether the world-centric representations were only based on objective variables of the agents or also dependent on sensory-motor signals such as gazing. This is a critical point for discussing world-centric representations of neurons. The analysis of Figure 9 seems insufficient to clarify the contributions of gazing. Firstly, the basic analyses about the relationships between the gazing point and the agent (self, prey, predator) position will be needed. How much does the gazing point correlate to the position of each agent? How often and with what types of behavior relations did the saccades occur? Secondly, neuronal tuning for only gazing variables should also be analyzed. (This kind of data is missing in Figure 9.) Although they

mentioned the influence of gazing and attention (Line 472-484), more thorough analysis and discussion seem to be needed.

We thank the reviewer for the productive suggestions regarding the question of how eye movements relate to our findings. Indeed, we think this is a valid concern for the reviewer to raise. The chief argument on this topic is that we observe uncorrelated (indeed, anti-correlated) maps for the three agents (self, prey, and other) – gaze could not account for all three of these. This to us is compelling evidence. Nonetheless, it is also possible to directly assess gaze and its influence on selectivity. We now add several new analyses.

- 1. The key concern seems to be that eye movements may correlate with the position of self/prey/predator in a world-centric (allocentric) manner and that neuronal coding for eye position may therefore appear, to our analysis, to be coding for self/prey/other. This is indeed a valid concern and we originally included, in an overly brief way, an analysis intended to address that concern. We now include a more complete analysis in the same vein, with similar results. Specifically, we show in our revised text that a GLM model that incorporates gaze position performs just as well as one that does not. In other words, gaze may contribute to modulation of firing rates (and likely does, see below), but these effects cannot explain our core mapping results because including gaze and allowing it to account for all variance does not reduce the strength of mapping. In other words, coding for gaze and for these mapping variables are largely independent.**
- 2. Addressing one of the reviewer's specific questions, we report that a significant proportion of neurons *are* tuned to gaze variables. However, this tuning is largely independent of positional tuning. Note that it is critical to fit the GLM model with both (gaze and position) types of variable using cross-validation to ensure that the significance of both variables are not inflated. Especially some relation between the gaze position and the agent position indicates that there can be inflation of gaze tuning once the agent relevant information is excluded (i.e. p-values would be higher when there is only one of the variables).**

In addition, the reviewer requests us to develop our analyses of gaze and eye position, above and beyond this concern. We appreciate having

the opportunity to do so. We now include the following three other new analyses.

3. We first asked how gaze is distributed relative to the positions of the self/prey/predator. We find that gaze moves around them quite a bit, and varies across the subjects (see numbers below). By and large subjects fixate the various items in the scene roughly equally, with most gaze towards the prey, and least towards the predator.
4. We asked how often subjects' eye behavior is described as either fixation, smooth pursuit, or saccade. We use position and velocity information to identify these behaviors. We find that gaze is mostly smooth pursuit (see numbers below).
5. We asked how the magnitude of eye movements relates to the complexity of the path of the prey. We hypothesized that shorter eye movements will accompany more complex paths. Instead of a saccade occurring during the complex trajectory, we expect there can be some time lead or lag of the saccade compared to the complex trajectory. Indeed, we found that there is a systematic relationship between the complexity of the prey's path (i.e. moment-to-moment curvature of the trajectory) and the magnitude of the eye-movements (measured in pixels). The large eye-movement leads the high curvature, which indicates the saccade (global scanning of the environment) happens before the pursuit becomes complex.

We add the following new text to the manuscript:

In the dynamic pursuit task, the position of the eyes are not fixed. Indeed, subjects continually scan the scene and follow specific items on screen. This leads to the possibility of a novel confound - specifically, our "world-centric" representations may come from simple gaze direction tuning.

We also offer some characterization of saccade behavior. One question related to its free eye movement is in what proportion of the time subjects foveate each agent (self, prey, and predator). To gain the broadest view, we analyzed 1-prey and 1-predator trials, wherein all types (subject, prey, and predator) of agent exist (7008 trials in two subjects). Our criterion for defining foveation was a range of 4 degree visual angle, which corresponded to 134 pixels (for reference, each agent was 60 pixel for width and height). We found that the foveation time was highest for the prey (31.85% and 19.05% for subjects K and P, respectively), and lowest for

the predator (19.09% and 5.11%, respectively) with a small proportion for foveating both agents (2.82% for subject K, 0.07% for subject P). Foveation on the subject was 24.34% (subject K) and 9.71% for subject P.

A second question is how eye movements relate to target pursuit. One possibility is that they maximally use fixations/ saccades. In other words, they may position their fovea around the predicted position of the target and fixate until the tracking target moves beyond some angle. An alternative possibility is to follow the target with smooth pursuit. (these two strategies have some heuristic relationship with discrete and continuous sampling, respectively). We found that the latter hypothesis corresponds to our data. Specifically, 90.88% (Subject K) and 94.70% (Subject P) of time bins have eye movements that are smooth pursuit, 4.03%/3.11% are saccadic fixation, and 5.10%/2.18% are movement.

We hypothesized that the appearance of the saccade was related to the complexity of pursuit trajectory. We used the curvature of the agent movement as a proxy for the complexity of trajectory, and measured cross-correlation between curvature value and magnitude of eye-movement. By this analysis, we wanted to answer two questions: (1) What is the nature of the saccade pattern? (Does large eye-movement correlate with large curvature/) and (2) Is there any systematic time lag between the saccade and the start of complex movement pattern? Indeed, we found that there is a systematic relationship between the complexity of the prey (i.e. moment-to-moment curvature of the trajectory) and the saccadic eye-movements (magnitude of the eye-movement pixel-wise). About 50% of the trials showed a significant cross-correlation between those two values (median correlation: $p=0.5020$, $n = 2052/3954$ trial for subject K; $p=0.5334$, $n=1395/3054$ trials for subject P; $p < 0.001$ by two-way binomial test for both subjects). The median time lag was -66.7 ms for subject K and -116.7 ms for subject P. The large eye-movement as leading the high curvature, which indicates the saccade (global scanning of the environment) happens before the pursuit becomes complex.

Finally, we asked whether eye position information influences the tuning of other variables. Specifically, we asked two questions: (1) whether the tuning for other non-gaze related variables is decreased when the gaze related variables are added, (2) whether the gaze variables are not significantly tuned when they are added to the fits because the correlated variables already captured variance. We found neither was the case: the tuning of other neurons remained significant even after adding gaze

position information (Figure 9). If the correlation really influenced the tuning, the tuning of prey position should be decreased for the largest amount as the foveation time was highest at the prey. However, tuning for the prey did not decrease significantly. In addition, a significant number of neurons are tuned to the gaze position ($n=10/37$, $p<0.001$ by two-way binomial test), which means there is variance still well captured by the gaze position information.

Figure S4. The time lead/lag correlation between complexity of the trajectory (using curvature of the trajectory as a proxy) and eye-movement magnitude. (A) The relationship between self-trajectory complexity and eye movement magnitude. The right panel is for subject K and the left panel is for subject P. (B) The relationship between prey trajectory and eye-movement magnitude.

Minor comments

1) Please provide the details of the analysis using spatial efficiency (SPAEF) in Methods.

We have now added the following new text to the methods. We apologize that this information did not appear earlier.

To compare the similarity between two positional filters, we used the SPAtial Efficiency metric (SPAEF). This is a mathematical technique that is derived from the geology literature, but has a much broader application. Formally, it allows for the quantification of the similarity of two two-dimensional filters with univariate scalars as entries. Prior literature suggests to be more robust than the 2D spatial correlation (Koch et al., 2018). It quantifies the similarity between two maps:

$$SPAEF = 1 - \sqrt{(A-1)^2 + (B-1)^2 - (C-1)^2}$$

A is the Pearson correlation between two maps, B is the ratio between the coefficients of variation for each map, and C is the activity similarity measured by histogram profiles. A zero SPAEF indicates orthogonal filters, whereas a positive SPAEF indicates similar filters and a negative SPAEF indicates anticorrelated filters.

2) Please provide the information on the experimental methods of eye-tracking in Methods.

We now include the following new text:

Eyetracking methods:

Subjects were rigidly head-fixed using a cranial implant that was located far from the eyes. Subjects were facing directly forwards with the eyes centered on the center of the computer monitor. Gaze position was measured with an Eyelink 1000 system at 1000 Hz. Calibration was performed daily for each subject using a 20-point calibration procedure in which they fixated briefly on a point displayed at random on a computer screen. Calibration quality was monitored throughout the recording session and was checked periodically during recording by pausing the task and running a calibration routine again. Careful post-hoc checks reveal little systematic error in tracking; subjects were found to be placing gaze on the targets. These tests concorded with casual observations made during and throughout testing by a trainee monitoring the subject and gaze at all times. For resulting analyses, the gaze x and y position were included in analyses.

3) Related to Figure 9. Please provide the information on what gazing variables were used in the analysis.

We now state clearly in the text that gaze x- and y-position at each time point was included.

4) Line 861~. What is the time window for fitting the spike trains to the model?

We apologize for not making this information clearer. This model, which was inspired by Hardcastle et al., 2017, does not assume a specific time window of events like the usual GLM approach in primate research (Park et al., 2014; Yates et al. 2017). Instead, like Hardcastle et al 2017, we used whole session time with ITI truncation as a big analysis unit since our pursuit task is not separately easy like other primate studies. The time bin used within that analytic unit was 16.67 ms. The following new text appears in the manuscript to highlight this point:

For neural analyses, we focused on the *whole trial epoch*, that is, the period from the time when all agents appear on the screen (trial start) until the end of the trial, defined as either (i) the time when the subject captures the prey, (ii) the time when the predator captures the subject, or (iii) 20 seconds pass without either other event occurring. Thus, the tuning maps presented below indicate the average firing of neurons (using 16.67 ms analysis bins) during the entire course of active behavior during the trial, rather than during any specific epoch.

Reviewer #2 (Remarks to the Author):

Overall, I found that this paper addressed an interesting problem. The introduction framed the experiments nicely within a broader ethological and anatomical framework. The task was well-designed with nice features and the data set was large. Analyses were detailed and thorough. I have only a few comments/ suggestions with one major comment.

We appreciate these positive comments. We have now addressed these comments; we believe that the changes have made the paper better. Detailed replies to the specific comments are found below.

Comments

1. The main thing I thought the study was missing was a detailed analysis of what the animal was actually doing. This could then form the basis of the neural analysis, which were descriptive and detailed, but could be more insightful. Perhaps the authors have tried some of these approaches. One possibility would be to analyze the direction of motion of the subject, moment by moment, and determine the extent to which it is driven more by pursuit of prey or avoidance of predators? This likely depends on the size of the reward, and the proximity of the predator. I realize this only occurs in a subset of trials. But it could be used to divide the neural activity into epochs, or look for some sort of switching phenomenon, as in switching between pursuit/avoidance. Also, how efficient was the prey pursuit? One could imagine something between a random walk and a predictive pursuit strategy, that tries to intersect the prey based on its current direction and speed.

The reviewer seems to believe that our analyses largely support our conclusions and adjudicate our hypotheses, but to think that our paper could benefit from additional analyses. We appreciate the opportunity to add new results to our paper. There is, of course, a fine balance here, between adding so much new material that it substantially changes our paper, and risks obscuring the key results - about the multicentric maps in dACC. In either case, we have added two major new analyses.

The reviewer's main interest seems to be on the question of how we can quantify the agent's strategy. We agree this is interesting and important. One straightforward way to do this is to divide behavior into pursuit (chasing) and flight (avoiding). There are, in turn, many ways to do this, but the most agnostic (model-free) way to do so is to calculate the distance between the agent, the prey, and predator, at each moment, as well as the

derivative of these two quantities. If the delta distance, dot product, and delta dot product is positive between subject and predator, for example, it will be more likely for the case of avoidance. Another example can be positive delta distance, positive dot product, and positive delta dot product would indicate that the subject is chasing the prey. Again, the chase can be reassured by the high value of the curvature for both subject and prey (predator has smaller curvature by design). This analysis in turn can identify avoidance, as we show below.

A

B

C

D

E

Figure S4. Transition between chase and evade behaviors in the prey pursuit task. Gray: self. Red: predator. Green: prey. (A) Trajectories of self, prey, and predator in an example trial. Red: distance between self and predator. Green: distance between self and prey. The filled circles indicate periods of avoidance, as detected by our algorithm. A cross indicates the chase periods and unfilled small circles indicate periods that are not successfully classified. The filled diamonds points are the location of each agent at the starting of the session. (B) distance between subject-prey and subject-predator over time in an example trial (same trial as in panel A). Filled circles indicate avoidance periods. (C) Derivative of distance as a function of time (same example trial as in panel B). (D) The dot product between the vector of self-movement and other agents, which is the key intermediate variable our algorithm uses to classify behavior. A value of -1 indicates that the movements of each agent have opposite directions (note the magnitude is converted to 1 as an unit vector), and 1 means the movement directions are aligned. (E) Derivative of dot product (i.e. of panel D).

We estimated the avoid and chase moments by basic statistics given each moment-by-moment (Supplementary figure 4). Included variables are change of the distance (delta distance: ‘getting close or far?’) and the dot product between the vectors of each agent (‘how similar the movements are’) and its change (‘are the agents moving similarly over time?’). For example, if the delta distance, dot product, and delta dot product is positive between subject and predator (example in figure), it will be more likely for the mode of avoidance. According to this method, clear avoidance times were 10.48% (subject K), 9.09% (subject H), and 12.99% (subject P) from the whole session.

A second analysis that the reviewer suggested was estimating the efficiency of the actual subject. In order to accomplish that, there should be some reference to compare. As the reviewer hinted, we decided to compare subjects’ behavior with that of simulated agents. It is described in the following new text:

Figure S5. (A) Illustration of algorithm of a newly simulated agent for estimating efficiency of subject's pursuing. The algorithm incorporates selective intake of the predator information depending on the distance (shaded red region) and predictive pursuit components, which is denoted as tau (Yoo et al., 2020). The physical inertia (yellow arrow), pursuit towards prey (cyan arrow), and avoid from predator (red arrow) are summed and resulted in single vector (green dashed arrow). (B) Percent of catching prey in 1-prey, 1-predator trials, where inefficient pursuit may end up being captured by predator. All subjects were between fully predictive agent and fully reactive agent with identical amount of tau parameters. (C) Percent of being captured by the predator within 20 seconds. The result in (B) and (C) may not sum to 1 because there are trials for time outs.

To more fully understand our subjects' behavior, we created multiple artificial agents that pursue prey and avoid predators. We used these agents to estimate the efficiency of the monkeys' observed algorithm (Figure S5). There are two components that vary across our five models: (1) the predictive parameter tau, which indicates how much extrapolation from previous Newtonian physics that an agent can make towards the future (Yoo et al., 2020), and (2) the distance-dependent influence of the predator. If the distance dependent function for the predator is large, it influences the subject even if the predator is far from it. All of these algorithms are bound to the same physical constraints, especially to a maximum speed and physical inertia. We divided the tau parameter into two categories (tau was either predictive or reactive) and the influence parameter into two categories (low distance / narrow or high distance / broad influence of predator). We then crosses these to

make four categories and added a final degenerate random walk model.

These models give a range of performances. The *random walk model* gives 0% capture of prey within 20 seconds (not surprisingly) (Figure S5 B). One trials with predators, actual subjects caught prey for 54.05% (subject K), and 38.97% (subject H). For this analysis, we also include data from a new subject whose data did not appear in our earlier submission (subject P), whose performance was between the other two (45.55%). Other agents that always predictively pursue (fixed tau value of 30 similar a subject result from Yoo et al., 2020) the prey yields a higher catch rate compared (86.54% for narrow attention for predator vs. 75.03% for broad attention for predator). However, if the agent is always reactive to pursue (fixed tau value of -30, which drastically differs from subject's average trajectory), then though the agent is faster than the prey, it rarely catches the prey. Instead, the probability of being captured by a predator increases (10.15% for narrow attention for predator, 0.13% for broad attention for predator). In conclusion, the predictive model with narrow attention is the most accurate descriptor.

2. A better figure indicating recording locations would be useful. Is this 9m? 24c? Medial area 6? Dorsal to the cingulate sulcus? This may be in the methods, but it would be useful to show locations and maybe given approximate AP coordinates.

We use the “standard” dACC used by our own lab in many other papers and by many other labs as well. This is a region adjacent to the corpus callosum, roughly rostral and caudal to the plane of the rostral genu. Like most other labs, our recordings favor the dorsal bank. The numbering of this region is disputed. Our lab has argued (based on connectivity) that this suprasulcal segment is anatomically continuous with the infrasulcal segment, and, like it, ought to be called area 24 (Heilbronner and Hayden, Annual Review in Neuroscience, 2016); however, Vogt has argued (based on cytoarchitecture) it does not constitute a single discrete area and should be thought of as a cingulate transition zone, and calls it 6/32, 8/32, and 9/32. We note that the general practice in the electrophysiological literature has been to treat it as a single area, and is thus more in line with our numbering approach than with Vogt's.

We also include the following new image which demonstrates our recording site:

3. The manuscript was well-organized but could definitely benefit from editing. Results are often presented in the present tense (are instead of were) and there are numerous typos.

We have done major edits for typos. As for the tense of the Results, this is a style preference and neither present nor past tense is considered incorrect by the journal.

Reviewers' Comments:

Reviewer #1:

Remarks to the Author:

My concerns have been addressed in the revised version of the manuscript. I do not have further comments. Now I would like to recommend the paper for publication.

Reviewer #2:

Remarks to the Author:

The authors have addressed my comments with additional analyses. I have no further comments.

Bruno Averbeck

Reviewer #1 (Remarks to the Author):

In this manuscript, Yoo et al. investigated neuronal coding for moving agents in the dorsal anterior cingulate cortex (dACC) of the monkeys. The authors trained the monkeys to perform the task in which the animals were required to control an avatar (self) to pursue prey and to escape from a predator on a monitor using a joystick. They found that a set of dACC neurons showed significant tuning for ‘world-centric’ variables such as the positions and velocities of the agents. Also, a set of dACC neurons showed significant tuning for ‘avatar-centric’ variables such as the distance and angle between the self and prey agents. The topic of this study is timely and interesting for readers with a broad background. However, there are one major and several minor concerns that should be addressed.

We appreciate these positive comments. We have now addressed all comments listed; we believe that the resulting changes have made the paper better. Detailed replies to the specific comments are found below.

Spatial coding of hippocampal place cells is thought to be ‘allocentric’ because activities of place cells are mostly tuned for the absolute coordinates of the environment and are independent of the sensory-motor information such as head-direction and locomotion speed. In the current study, the authors claimed that the populations of dACC neurons represent ‘world-centric’ spatial representations of the agents. Here, the ‘world-centric’ is used as the similar meaning of ‘allocentric.’

Indeed, the reviewer is correct about this. We apologize for any confusion, and have added the recommended terminology in hopes of making things less confusing. Specifically we now explicitly add the terms “allocentric” and “egocentric” as comparators in the Introduction. However, we still preferentially use the original terms throughout so as to not to imply that these neurons are tested using a standard maze/navigation-like task.

However, they did not show clearly whether the world-centric representations were only based on objective variables of the agents or also dependent on sensory-motor signals such as gazing. This is a critical point for discussing world-centric representations of neurons. The analysis of Figure 9 seems insufficient to clarify the contributions of gazing. Firstly, the basic analyses about the relationships between the gazing point and the agent (self, prey, predator) position will be needed. How much does the gazing point correlate to the position of each agent? How often and with what types of behavior relations did the saccades occur? Secondly, neuronal tuning for only gazing variables should also be analyzed. (This kind of data is missing in Figure 9.) Although they men-

tioned the influence of gazing and attention (Line 472-484), more thorough analysis and discussion seem to be needed.

We thank the reviewer for the productive suggestions regarding the question of how eye movements relate to our findings. Indeed, we think this is a valid concern for the reviewer to raise. The chief argument on this topic is that we observe uncorrelated (indeed, anti-correlated) maps for the three agents (self, prey, and other) – gaze could not account for all three of these. This to us is compelling evidence. Nonetheless, it is also possible to directly assess gaze and its influence on selectivity. We now add several new analyses.

- 1. The key concern seems to be that eye movements may correlate with the position of self/prey/predator in a world-centric (allocentric) manner and that neuronal coding for eye position may therefore appear, to our analysis, to be coding for self/prey/other. This is indeed a valid concern and we originally included, in an overly brief way, an analysis intended to address that concern. We now include a more complete analysis in the same vein, with similar results. Specifically, we show in our revised text that a GLM model that incorporates gaze position performs just as well as one that does not. In other words, gaze may contribute to modulation of firing rates (and likely does, see below), but these effects cannot explain our core mapping results because including gaze and allowing it to account for all variance does not reduce the strength of mapping. In other words, coding for gaze and for these mapping variables are largely independent.**
- 2. Addressing one of the reviewer’s specific questions, we report that a significant proportion of neurons *are* tuned to gaze variables. However, this tuning is largely independent of positional tuning. Note that it is critical to fit the GLM model with both (gaze and position) types of variable using cross-validation to ensure that the significance of both variables are not inflated. Especially some relation between the gaze position and the agent position indicates that there can be inflation of gaze tuning once the agent relevant information is excluded (i.e. p-values would be higher when there is only one of the variables).**

In addition, the reviewer requests us to develop our analyses of gaze and eye position, above and beyond this concern. We appreciate having

the opportunity to do so. We now include the following three other new analyses.

3. We first asked how gaze is distributed relative to the positions of the self/prey/predator. We find that gaze moves around them quite a bit, and varies across the subjects (see numbers below). By and large subjects fixate the various items in the scene roughly equally, with most gaze towards the prey, and least towards the predator.
4. We asked how often subjects' eye behavior is described as either fixation, smooth pursuit, or saccade. We use position and velocity information to identify these behaviors. We find that gaze is mostly smooth pursuit (see numbers below).
5. We asked how the magnitude of eye movements relates to the complexity of the path of the prey. We hypothesized that shorter eye movements will accompany more complex paths. Instead of a saccade occurring during the complex trajectory, we expect there can be some time lead or lag of the saccade compared to the complex trajectory. Indeed, we found that there is a systematic relationship between the complexity of the prey's path (i.e. moment-to-moment curvature of the trajectory) and the magnitude of the eye-movements (measured in pixels). The large eye-movement leads the high curvature, which indicates the saccade (global scanning of the environment) happens before the pursuit becomes complex.

We add the following new text to the manuscript:

In the dynamic pursuit task, the position of the eyes are not fixed. Indeed, subjects continually scan the scene and follow specific items on screen. This leads to the possibility of a novel confound - specifically, our “world-centric” representations may come from simple gaze direction tuning.

We also offer some characterization of saccade behavior. One question related to its free eye movement is in what proportion of the time subjects foveate each agent (self, prey, and predator). To gain the broadest view, we analyzed 1-prey and 1-predator trials, wherein all types (subject, prey, and predator) of agent exist (7008 trials in two subjects). Our criterion for defining foveation was a range of 4 degree visual angle, which corresponded to 134 pixels (for reference, each agent was 60 pixel for width and height). We found that the foveation time was highest for the prey (31.85% and 19.05% for subjects K and P, respectively), and lowest for the predator (19.09% and

5.11%, respectively) with a small proportion for foveating both agents (2.82% for subject K, 0.07% for subject P). Foveation on the subject was 24.34% (subject K) and 9.71% for subject P.

A second question is how eye movements relate to target pursuit. One possibility is that they maximally use fixations/ saccades. In other words, they may position their fovea around the predicted position of the target and fixate until the tracking target moves beyond some angle. An alternative possibility is to follow the target with smooth pursuit. (these two strategies have some heuristic relationship with discrete and continuous sampling, respectively). We found that the latter hypothesis corresponds to our data. Specifically, 90.88% (Subject K) and 94.70% (Subject P) of time bins have eye movements that are smooth pursuit, 4.03%/3.11% are saccadic fixation, and 5.10%/2.18% are movement.

We hypothesized that the appearance of the saccade was related to the complexity of pursuit trajectory. We used the curvature of the agent movement as a proxy for the complexity of trajectory, and measured cross-correlation between curvature value and magnitude of eye-movement. By this analysis, we wanted to answer two questions: (1) What is the nature of the saccade pattern? (Does large eye-movement correlate with large curvature/) and (2) Is there any systematic time lag between the saccade and the start of complex movement pattern? Indeed, we found that there is a systematic relationship between the complexity of the prey (i.e. moment-to-moment curvature of the trajectory) and the saccadic eye-movements (magnitude of the eye-movement pixel-wise). About 50% of the trials showed a significant cross-correlation between those two values (median correlation: $p=0.5020$, $n = 2052/3954$ trial for subject K; $p=0.5334$, $n=1395/3054$ trials for subject P; $p < 0.001$ by two-way binomial test for both subjects). The median time lag was -66.7 ms for subject K and -116.7 ms for subject P. The large eye-movement as leading the high curvature, which indicates the saccade (global scanning of the environment) happens before the pursuit becomes complex.

Finally, we asked whether eye position information influences the tuning of other variables. Specifically, we asked two questions: (1) whether the tuning for other non-gaze related variables is decreased when the gaze related variables are added, (2) whether the gaze variables are not significantly tuned when they are added to the fits because the correlated variables already captured variance. We found neither was the case: the tuning of other neurons remained significant even after adding gaze position in-

formation (Figure 9). If the correlation really influenced the tuning, the tuning of prey position should be decreased for the largest amount as the foveation time was highest at the prey. However, tuning for the prey did not decrease significantly. In addition, a significant number of neurons are tuned to the gaze position ($n=10/37$, $p<0.001$ by two-way binomial test), which means there is variance still well captured by the gaze position information.

Figure S4. The time lead/lag correlation between complexity of the trajectory (using curvature of the trajectory as a proxy) and eye-movement magnitude. (A) The relationship between self-trajectory complexity and eye movement magnitude. The right panel is for subject K and the left panel is for subject P. (B) The relationship between prey trajectory and eye-movement magnitude.

Minor comments

1) Please provide the details of the analysis using spatial efficiency (SPAEF) in Methods.

We have now added the following new text to the methods. We apologize that this information did not appear earlier.

To compare the similarity between two positional filters, we used the Spatial Efficiency metric (SPAEF). This is a mathematical technique that is derived from the geology literature, but has a much broader application. Formally, it allows for the quantification of the similarity of two two-dimensional filters with univariate scalars as entries. Prior literature suggests to be more robust than the 2D spatial correlation (Koch et al., 2018). It quantifies the similarity between two maps:

$$SPAEF = 1 - \sqrt{(A-1)^2 + (B-1)^2 - (C-1)^2}$$

A is the Pearson correlation between two maps, B is the ratio between the coefficients of variation for each map, and C is the activity similarity measured by histogram profiles. A zero SPAEF indicates orthogonal filters, whereas a positive SPAEF indicates similar filters and a negative SPAEF indicates anticorrelated filters.

2) Please provide the information on the experimental methods of eye-tracking in Methods.

We now include the following new text:

Eyetracking methods:

Subjects were rigidly head-fixed using a cranial implant that was located far from the eyes. Subjects were facing directly forwards with the eyes centered on the center of the computer monitor. Gaze position was measured with an Eyelink 1000 system at 1000 Hz. Calibration was performed daily for each subject using a 20-point calibration procedure in which they fixated briefly on a point displayed at random on a computer screen. Calibration quality was monitored throughout the recording session and was checked periodically during recording by pausing the task and running a calibration routine again. Careful post-hoc checks reveal little systematic error in tracking; subjects were found to be placing gaze on the targets. These tests concorded with casual observations made during and throughout testing by a trainee monitoring the subject and gaze at all times. For resulting analyses, the gaze x and y position were included in analyses.

3) Related to Figure 9. Please provide the information on what gazing variables were used in the analysis.

We now state clearly in the text that gaze x- and y-position at each time point was included.

4) Line 861~. What is the time window for fitting the spike trains to the model?

We apologize for not making this information clearer. This model, which was inspired by Hardcastle et al., 2017, does not assume a specific time window of events like the usual GLM approach in primate research (Park et al., 2014; Yates et al. 2017). Instead, like Hardcastle et al 2017, we used whole session time with ITI truncation as a big analysis unit since our pursuit task is not separately easy like other primate studies. The time bin used within that analytic unit was 16.67 ms. The following new text appears in the manuscript to highlight this point:

For neural analyses, we focused on the *whole trial epoch*, that is, the period from the time when all agents appear on the screen (trial start) until the end of the trial, defined as either (i) the time when the subject captures the prey, (ii) the time when the predator captures the subject, or (iii) 20 seconds pass without either other event occurring. Thus, the tuning maps presented below indicate the average firing of neurons (using 16.67 ms analysis bins) during the entire course of active behavior during the trial, rather than during any specific epoch.

Reviewer #2 (Remarks to the Author):

Overall, I found that this paper addressed an interesting problem. The introduction framed the experiments nicely within a broader ethological and anatomical framework. The task was well-designed with nice features and the data set was large. Analyses were detailed and thorough. I have only a few comments/ suggestions with one major comment.

We appreciate these positive comments. We have now addressed these comments; we believe that the changes have made the paper better. Detailed replies to the specific comments are found below.

Comments

1. The main thing I thought the study was missing was a detailed analysis of what the animal was actually doing. This could then form the basis of the neural analysis, which were descriptive and detailed, but could be more insightful. Perhaps the authors have tried some of these approaches. One possibility would be to analyze the direction of motion of the subject, moment by moment, and determine the extent to which it is driven more by pursuit of prey or avoidance of predators? This likely depends on the size of the reward, and the proximity of the predator. I realize this only occurs in a subset of trials. But it could be used to divide the neural activity into epochs, or look for some sort of switching phenomenon, as in switching between pursuit/avoidance. Also, how efficient was the prey pursuit? One could imagine something between a random walk and a predictive pursuit strategy, that tries to intersect the prey based on its current direction and speed.

The reviewer seems to believe that our analyses largely support our conclusions and adjudicate our hypotheses, but to think that our paper could benefit from additional analyses. We appreciate the opportunity to add new results to our paper. There is, of course, a fine balance here, between adding so much new material that it substantially changes our paper, and risks obscuring the key results - about the multicentric maps in dACC. In either case, we have added two major new analyses.

The reviewer's main interest seems to be on the question of how we can quantify the agent's strategy. We agree this is interesting and important. One straightforward way to do this is to divide behavior into pursuit (chasing) and flight (avoiding). There are, in turn, many ways to do this, but the most agnostic (model-free) way to do so is to calculate the distance between the agent, the prey, and predator, at each moment, as well as the de-

rivative of these two quantities. If the delta distance, dot product, and delta dot product is positive between subject and predator, for example, it will be more likely for the case of avoidance. Another example can be positive delta distance, positive dot product, and positive delta dot product would indicate that the subject is chasing the prey. Again, the chase can be reassured by the high value of the curvature for both subject and prey (predator has smaller curvature by design). This analysis in turn can identify avoidance, as we show below.

A

B

C

D

E

Figure S4. Transition between chase and evade behaviors in the prey pursuit task. Gray: self. Red: predator. Green: prey. (A) Trajectories of self, prey, and predator in an example trial. Red: distance between self and predator. Green: distance between self and prey. The filled circles indicate periods of avoidance, as detected by our algorithm. A cross indicates the chase periods and unfilled small circles indicate periods that are not successfully classified. The filled diamonds points are the location of each agent at the starting of the session. (B) distance between subject-prey and subject-predator over time in an example trial (same trial as in panel A). Filled circles indicate avoidance periods. (C) Derivative of distance as a function of time (same example trial as in panel B). (D) The dot product between the vector of self-movement and other agents, which is the key intermediate variable our algorithm uses to classify behavior. A value of -1 indicates that the movements of each agent have opposite directions (note the magnitude is converted to 1 as an unit vector), and 1 means the movement directions are aligned. (E) Derivative of dot product (i.e. of panel D).

We estimated the avoid and chase moments by basic statistics given each moment-by-moment (Supplementary figure 4). Included variables are change of the distance (delta distance: ‘getting close or far?’) and the dot product between the vectors of each agent (‘how similar the movements are’) and its change (‘are the agents moving similarly over time?’). For example, if the delta distance, dot product, and delta dot product is positive between subject and predator (example in figure), it will be more likely for the mode of avoidance. According to this method, clear avoidance times were 10.48% (subject K), 9.09% (subject H), and 12.99% (subject P) from the whole session.

A second analysis that the reviewer suggested was estimating the efficiency of the actual subject. In order to accomplish that, there should be some reference to compare. As the reviewer hinted, we decided to compare subjects’ behavior with that of simulated agents. It is described in the following new text:

Figure S5. (A) Illustration of algorithm of a newly simulated agent for estimating efficiency of subject's pursuing. The algorithm incorporates selective intake of the predator information depending on the distance (shaded red region) and predictive pursuit components, which is denoted as tau (Yoo et al., 2020). The physical inertia (yellow arrow), pursuit towards prey (cyan arrow), and avoid from predator (red arrow) are summed and resulted in single vector (green dashed arrow). (B) Percent of catching prey in 1-prey, 1-predator trials, where inefficient pursuit may end up being captured by predator. All subjects were between fully predictive agent and fully reactive agent with identical amount of tau parameters. (C) Percent of being captured by the predator within 20 seconds. The result in (B) and (C) may not sum to 1 because there are trials for time outs.

To more fully understand our subjects' behavior, we created multiple artificial agents that pursue prey and avoid predators. We used these agents to estimate the efficiency of the monkeys' observed algorithm (Figure S5). There are two components that vary across our five models: (1) the predictive parameter tau, which indicates how much extrapolation from previous Newtonian physics that an agent can make towards the future (Yoo et al., 2020), and (2) the distance-dependent influence of the predator. If the distance dependent function for the predator is large, it influences the subject even if the predator is far from it. All of these algorithms are bound to the same physical constraints, especially to a maximum speed and physical inertia. We divided the tau parameter into two categories (tau was either predictive or reactive) and the influence parameter into two categories (low distance / narrow or high distance / broad influence of predator). We then crosses these to make four categories and added a final degenerate random walk model.

These models give a range of performances. The *random walk model* gives 0% capture of prey within 20 seconds (not surprisingly) (Figure S5 B). One trials with predators, actual subjects caught prey for 54.05% (subject K), and 38.97% (subject H). For this analysis, we also include data from a new subject whose data did not appear in our earlier submission (subject P), whose performance was between the other two (45.55%). Other agents that always predictively pursue (fixed tau value of 30 similar a subject result from Yoo et al., 2020) the prey yields a higher catch rate compared (86.54% for narrow attention for predator vs. 75.03% for broad attention for predator). However, if the agent is always reactive to pursue (fixed tau value of -30, which drastically differs from subject's average trajectory), then though the agent is faster than the prey, it rarely catches the prey. Instead, the probability of being captured by a predator increases (10.15% for narrow attention for predator, 0.13% for broad attention for predator). In conclusion, the predictive model with narrow attention is the most accurate descriptor.

2. A better figure indicating recording locations would be useful. Is this 9m? 24c? Medial area 6? Dorsal to the cingulate sulcus? This may be in the methods, but it would be useful to show locations and maybe given approximate AP coordinates.

We use the “standard” dACC used by our own lab in many other papers and by many other labs as well. This is a region adjacent to the corpus callosum, roughly rostral and caudal to the plane of the rostral genu. Like most other labs, our recordings favor the dorsal bank. The numbering of this region is disputed. Our lab has argued (based on connectivity) that this suprasulcal segment is anatomically continuous with the infrasulcal segment, and, like it, ought to be called area 24 (Heilbronner and Hayden, Annual Review in Neuroscience, 2016); however, Vogt has argued (based on cytoarchitecture) it does not constitute a single discrete area and should be thought of as a cingulate transition zone, and calls it 6/32, 8/32, and 9/32. We note that the general practice in the electrophysiological literature has been to treat it as a single area, and is thus more in line with our numbering approach than with Vogt's.

We also include the following new image which demonstrates our recording site:

3. The manuscript was well-organized but could definitely benefit from editing. Results are often presented in the present tense (are instead of were) and there are numerous typos.

We have done major edits for typos. As for the tense of the Results, this is a style preference and neither present nor past tense is considered incorrect by the journal.